# Hierarchical Causal Abduction: A Foundation Framework for Explainable Model Predictive Control

**Ramesh Arvind Naagarajan** [1]  **Zühal Wagner** [1]  **Stefan Streif** [1 2]

## Abstract

Model Predictive Control (MPC) is widely used to operate safety-critical infrastructure by predicting future trajectories and optimizing control actions. However, nonlinear dynamics, hard safety constraints, and numerical optimization often render individual control moves opaque to human operators, undermining trust and hindering deployment. This paper presents Hierarchical Causal Abduction (HCA), which combines (i) physics-informed reasoning via domain knowledge graphs, (ii) optimization evidence from Karush–Kuhn–Tucker (KKT) multipliers, and (iii) temporal causal discovery via the PCMCI algorithm to generate faithful, human-interpretable explanations for control actions computed by nonlinear MPC. Across three diverse control applications (greenhouse climate, building HVAC, chemical process engineering) with expert validation, HCA improves explanation accuracy by 53% over LIME (0.478 vs. 0.311) using a single set of cross-domain parameters without per-domain tuning; domain-specific KKT-threshold calibration over 2–3 days further increases accuracy to 0.88. Ablation studies confirm that each evidence source is essential, with 32–37% accuracy degradation when any component is removed, and HCA's ranking-and-validation methodology generalizes beyond MPC to other prediction-based decision systems, including learning-based control and trajectory planning.

## 1. Introduction

Model Predictive Control (MPC) optimizes system operation by reasoning over the predicted system behavior over future horizons, enabling proactive and efficient control of critical infrastructure (Wu et al., 2025). However, MPC's prediction-based nature creates an explainability challenge: standard Explainable AI (XAI) methods fail to explain temporal causality, the phenomenon where current actions are determined by anticipated future violations of *system constraints* rather than the current state of the system (Chou et al., 2022; Carloni et al., 2025; Hettikankanamage et al., 2025). While a traditional reactive controller waits until a system constraint is actually violated, a predictive *greenhouse* controller may preemptively activate the *cooling* system while the current temperature is still within bounds, because predicted future solar radiation indicates that temperature constraints will be exceeded hours later. This *temporal causality*, where actions are determined by predicted future states, is a defining characteristic of MPC and systems with *accessible optimization objectives and constraints*. HCA requires explicit access to the optimization problem and constraints; applicability to model-free RL and black-box learned controllers remains future work.

As a concrete example, consider a greenhouse controller that activates ventilation to 60% when the current temperature is 23°C (below the 25°C limit) and humidity is 78% (below 85%). Why? HCA reveals that the optimizer is preventing a *predicted* humidity violation at $t+3\,$h from anticipated transpiration increase, a temporal causal chain invisible without analyzing the optimization trajectory. In our expert evaluation, domain experts could not correctly identify the causal mechanism behind MPC actions in 62% of cases, even with full model access.

Current XAI methods (LIME, SHAP (Ribeiro et al., 2016; Lundberg & Lee, 2017)) focus on instantaneous feature importance and cannot explain this temporal reasoning; they are designed for reactive systems, not predictive ones. MPC decisions require reasoning about interventions *"What will happen if we apply action $u_t$ now?"* and counterfactuals *"Would a violation occur without this action?"* (Pearl, 2009; 2018, pp. 70–106, pp. 1–42) which standard XAI methods do not expose. This gap motivates a specialized framework

[1]Professorship for Automatic Control and System Dynamics, Chemnitz University of Technology, Chemnitz, Germany [2]Department of Bioresources, Fraunhofer Institute for Molecular Biology and Applied Ecology, Giessen, Germany. Correspondence to: Stefan Streif <stefan.streif@etit.tu-chemnitz.de>.

*Proceedings of the 43$^{rd}$ International Conference on Machine Learning*, Seoul, South Korea. PMLR 306, 2026. Copyright 2026 by the author(s).

for explaining decisions where the causal pathway involves predicted future states and optimization over multi-step horizons (Runge, 2019; Heaton & Wu Fung, 2023; Carloni et al., 2025).

In this work, we introduce Hierarchical Causal Abduction (HCA), a framework that integrates optimization rigor (KKT multipliers), physical relationships (knowledge graphs), and temporal causal discovery (PCMCI) to expose *why* MPC acted (constraint necessity via counterfactuals), *what* physical mechanisms drove the decision (causal graph reasoning), and *when* temporal dependencies triggered the action (lagged causal discovery). The primary contribution is a principled hypothesis-ranking and counterfactual-validation architecture that fuses these three evidence sources into a single coherent explanation.

**Scope and positioning.** HCA applies techniques from the ML and XAI literature (causal discovery, knowledge-graph reasoning, counterfactual analysis) to explain automated decisions in *optimization-based control systems*. This is distinct from the predominant XAI focus on explaining learned model internals (DNNs, LLMs, ConvNets), though the underlying question, "why did the system choose *this* action *now*?", is shared with model-based RL, planning, and neural MPC. We deliberately formalize and evaluate the framework where ground-truth causal verification is possible (the optimization problem is accessible and the KKT system provides a true oracle), and discuss the bridge to learned controllers in Sec. 5.3 and Sec. 6.

Three specific contributions are made: (1) a unified explanation framework for predictive control systems with accessible optimization formulations; (2) empirical validation across three diverse domains (greenhouse, building automation HVAC, chemical process engineering), where HCA improves explanation accuracy by 53% over LIME (0.478 vs. 0.311) using transferable parameters; and (3) ablation studies confirming that all evidence sources are essential, with 32–37% accuracy degradation when any component is removed. With minimal domain-specific calibration (2-3 days), Answer Correctness (AC) further improves to approximately 0.88, demonstrating both cross-domain generalizability and practical deployability.

## 2. Related Work

**XAI Methods and MPC Analysis:** Foundational feature attribution methods like LIME and SHAP (Ribeiro et al., 2016; Lundberg & Lee, 2017) are designed for static inputs. In the context of MPC, they typically attribute decisions solely to the current state, ignoring the underlying optimization logic and temporal feedback. Even recent adaptations for learning-based MPC often treat the controller as a black box (Utama et al., 2022b). Recent surveys confirm the growing need for MPC-specific explainability (Wu et al., 2025); Schneider et al. (Utama et al., 2022a) apply SHAP to MPC but still treat the controller as input–output, missing constraint-driven reasoning. Classical MPC interpretability relies on KKT multipliers for mathematical rigor (Boyd & Vandenberghe, 2004; Berkenkamp et al., 2016; Nocedal & Wright, 2006, pp. 243–244), but these are not always easy for operators to interpret in terms of physical system behavior.

**Explanation Requirements in MPC vs. RL:** MPC and RL both optimize sequential decisions, but their explanation primitives differ (Bertsekas, 2019). MPC solves an explicit constrained optimization at every step: the objective, dynamics, and constraints are accessible, and KKT multipliers expose which constraints bind, with what marginal cost. RL learns an implicit policy or value function in which constraints (if any) are folded into reward shaping or learned penalties. Reward-decomposition methods (Juozapaitis et al., 2019; Rietz et al., 2022) attribute decisions to component value functions but cannot point to a binding constraint, because none is exposed. In this regard, MPC explanation must contend with predicted-future-violation reasoning that static-input XAI methods (LIME, SHAP) cannot represent. HCA exploits MPC's accessible optimization to provide a verifiable oracle for explanation correctness: when HCA attributes an action to constraint $i^*$ at horizon step $t+k$, this is checked by relaxing that constraint and resolving (Prop. C.2). In the context of black-box / model-free controllers no such oracle is available, and rigorous evaluation of the explanation itself becomes ambiguous; we view starting where the oracle exists as a methodological strength and discuss extensions to learned controllers (model-based RL, neural MPC) in Sec. 5.3.

**Physics-Informed Neural Networks (PINNs):** PINNs have been applied to control for state estimation and surrogate modeling (Antonelo et al., 2024; Nicodemus et al., 2022). While they enforce physical consistency and capture system dynamics, they function primarily as predictive models rather than decision explainers. Crucially, they do not explicitly expose the optimization rationale, such as active constraints or the specific actions that predicted violations necessitate.

**Inverse Optimal Control (IOC):** IOC methods recover underlying cost functions from observed trajectories to interpret agent behavior (Mombaur et al., 2010; Porcari et al., 2025). However, these global explanations emphasize objectives rather than constraints and typically fail to reveal when safety limits or actuator bounds drive MPC decisions.

**Template-Based Narrative Generation:** Template-based systems summarize MPC objectives and active constraints (Naagarajan & Streif, 2025) but often omit causal justifications and optimization trade-offs, yielding static

descriptions rather than context-sensitive abductive explanations.

**Causal Inference and Causal XAI:** Causal inference and causal XAI use structural causal models to generate counterfactual explanations (Pearl, 2009; Peters et al., 2017; Schölkopf et al., 2021; Holzinger et al., 2020) but are mainly designed for static prediction tasks and do not model MPC's constrained finite-horizon optimization with endogenous control inputs.

**Temporal Causal Discovery:** Temporal causal discovery methods such as PCMCI uncover lagged nonlinear dependencies in time series and can handle high-dimensional, non-stationary data (Runge, 2019; Balsells-Rodas et al., 2021; Carloni et al., 2025). However, they operate on passively observed trajectories and ignore the internal optimization logic of MPC, so they cannot explain *why* a particular control action was chosen at a given time.

**Knowledge Graphs in Control:** Domain-specific knowledge graphs encode physical components, dynamics, and constraints to support diagnosis (Pan et al., 2024; Chen et al., 2019; Liu et al., 2025). However, they typically operate independently of the controller's optimization objectives and ignore active constraints or forecasted disturbances.

**Gap:** Prior approaches address parts of the MPC explanation problem: (a) feature-attribution methods (LIME, SHAP) explain static mappings but miss temporal optimization logic; (b) causal discovery methods (PCMCI) identify temporal dependencies but ignore the optimizer's internal reasoning; (c) knowledge-graph approaches encode physical mechanisms but cannot determine which ones are active at a given timestep. No prior work unifies all three.

# 3. Methods

Given an optimal MPC action $u_k^*$, HCA generates an explanation in five steps: **(1)** extract KKT multipliers $\lambda_i$ to identify active constraints and rank them by influence on cost; **(2)** traverse the domain knowledge graph $G_{KG}$ to recover physical causal pathways linking disturbances, states, and controls; **(3)** query the PCMCI-learned causal graph $G_c$ for temporal precedent, checking whether parent variables showed anomalous deviations at their discovered lags; **(4)** evaluate candidate hypotheses in operational priority order (safety $\rightarrow$ optimization $\rightarrow$ prediction $\rightarrow$ economics $\rightarrow$ history), accepting the first hypothesis supported by evidence and validating it via counterfactual MPC re-solve; and **(5)** assemble the structured evidence into a human-readable explanation via LLM synthesis. The remainder of this section formalizes each step; Algorithm 1 and Figure 1 provide the complete pipeline.

## 3.1. Problem Formulation

A discrete-time nonlinear optimal control problem over a finite horizon $H \in \mathbb{N}$ is considered. At a decision instant, the controller is given the measured state $x_{\text{meas}}$ and a forecast of external disturbances $\{\hat{d}_k\}_{k=0}^{H-1}$, and computes an input sequence $\{u_k\}_{k=0}^{H-1}$ by solving with stage cost $\ell$, terminal cost $\ell_T$, dynamics $f$, and inequality constraints $g, g_T$. Problem (1) is a standard nonlinear MPC formulation (DQ, 2009; Berkenkamp et al., 2016); HCA is not restricted to this particular setup and applies to other MPC variants as long as the underlying optimal control problem and its constraints are accessible.

In Model Predictive Control, problem (1) is solved in receding-horizon fashion: at each sampling instant, it is initialized with $x_{\text{meas}}$ and $\{\hat{d}_k\}_{k=0}^{H-1}$, only $u_0^*$ is applied, and the horizon is shifted forward. The analysis targets *open-loop optimal control decisions at each discrete time step*, based solely on the state and disturbance forecast available at decision time. The complete nonlinear MPC formulation for the greenhouse application appears in Appendix J.

$$
\begin{aligned}
\min_{\{u_k\}_{k=0}^{H-1}} \quad & \sum_{k=0}^{H-1} \ell(x_k, u_k) + \ell_T(x_H) \\
\text{s.t.} \quad & x_0 = x_{\text{meas}}, \\
& x_{k+1} = f(x_k, u_k, \hat{d}_k), \quad k = 0, \ldots, H-1, \\
& g(x_k, u_k, \hat{d}_k) \leq 0, \quad k = 0, \ldots, H-1, \\
& g_T(x_H) \leq 0,
\end{aligned}
\tag{1}
$$

## 3.2. Temporal Causality in MPC

Traditional XAI typically analyzes static predictive models of the form $y_k = f(x_k)$, where $x_k$ denotes features at time step $k$ and $y_k$ the corresponding prediction. Such models focus on how features at a single time instant influence an instantaneous output and do not explicitly model how current decisions affect future trajectories or constraint satisfaction.

In MPC, by contrast, the control input $u_k$ is chosen by optimizing predicted future states $\{x_{k+j}\}_{j=0}^{H}$ over a finite horizon. Actions can therefore be primarily driven by anticipated future violations. For example, in a greenhouse with an upper temperature limit $T^{\max} = 25°C$, the current temperature may be $T_k = 22°C$, yet the prediction without cooling yields $T_{k+3} = 28°C > T^{\max}$ due to forecast solar radiation. An MPC controller activates cooling already at time $k$ to ensure $g(x_{k+j}, u_{k+j}, \hat{d}_{k+j}) \leq 0$ for some $j \in \{1, \ldots, H\}$; a purely reactive controller would wait until $T_k$ itself exceeds $T^{\max}$.

HCA is designed to make this temporal causality explicit. For each optimal input $u_k^*$, HCA generates candidate ex-

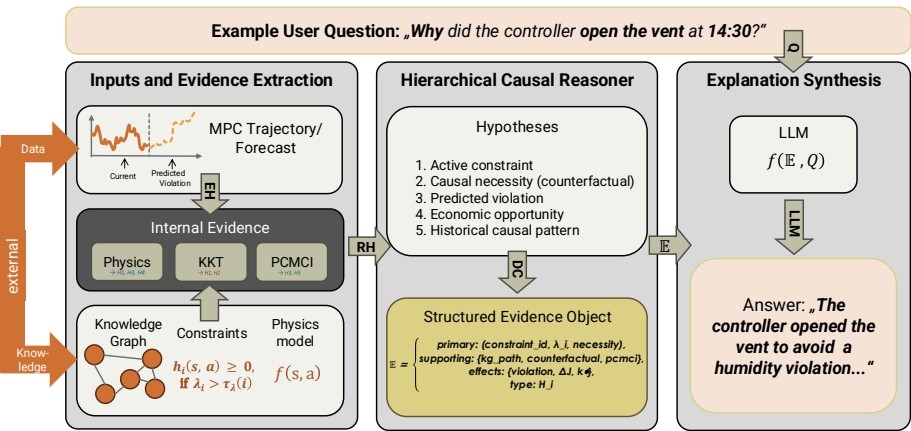

*Figure 1.* HCA workflow: three evidence sources (Physics, KKT, PCMCI) feed the Hierarchical Causal Reasoner (EH = Evaluate Hypothesis, RH = Rank Hypotheses, DC = Get Deeper Context), which is then synthesized by an LLM into natural-language explanations.

planations for each action, ranks them using its evidence sources, and validates the top-ranked ones via counterfactual MPC re-solves; see Sections 3.3–3.5, Figure 1, and Algorithm 1.

### 3.3. Evidence Source 1: Physics-Informed Reasoning

A domain knowledge graph $G_{KG}$ encodes qualitative physical relationships as nodes (states, controls, disturbances) and directed edges $x \to y$ labeled with a sign $\sigma \in \{+, -\}$, indicating whether an increase in $x$ tends to increase $(+)$ or decrease $(-)$ $y$ under nominal operating conditions. The framework is domain-general: any control system whose physics can be expressed as signed causal influences admits a $G_{KG}$. The graph was validated by 3 domain experts $(\kappa = 0.89)$.

During explanation, HCA traverses $G_{KG}$ downstream (from disturbances and actions to constraints) and upstream (from active constraints to potential drivers) to assemble physically plausible causal chains (details in Appendix E).

**Running Example (Greenhouse):** The graph $G_{KG}$ is constructed from the greenhouse climate and crop models in Sathyanarayanan et al. (2024); each edge reflects a term in the governing equations, with its sign indicating the local effect of that term. For instance, solar radiation has a positive effect on temperature $(Q_{rad} \xrightarrow{+} T)$, while ventilation decreases both temperature and humidity $(u_V \xrightarrow{-} T,\ u_V \xrightarrow{-} H)$. This enables interpretable statements such as: *"ventilation increased because forecast humidity and $CO_2$ levels would otherwise exceed their upper bounds."*

### 3.4. Evidence Source 2: Optimization-Based Analysis

At each time step, the MPC problem (1) is solved by a Nonlinear Program (NLP) solver, returning KKT multipliers $\lambda_i \geq 0$ for each inequality constraint $g_i$. These quantify the cost reduction if a constraint were relaxed; by KKT theory, $\lambda_i > 0$ only if the constraint is active at optimality.

For hard-constrained systems, thresholds $\tau_\lambda(i)$ identify active constraints from solver-returned multipliers, achieving 96–98% classification accuracy (Appendix G). For soft-constrained systems with penalty-based comfort bands, explicit KKT multipliers do not exist for comfort-band constraints (which are enforced via soft penalties in $\ell$); constraint activity is instead identified via counterfactual analysis (Appendix G.2).

**Running Example (Greenhouse):** The greenhouse uses soft penalty-based comfort bands, so constraint activity for temperature/humidity/$CO_2$ comfort is identified via counterfactual re-solve rather than nonzero $\lambda_i$. Building HVAC and TEP, by contrast, use hard solver-returned multipliers.

HCA uses two distinct counterfactuals (CF) that address complementary questions.

**(CF-A) Action removal** ($\mathtt{SIM}(u{=}0)$, Algorithm 2 in App. B.2): force $u(t) = 0$ at the current step, but let the optimizer re-optimize $u(t{+}1), \ldots, u(t{+}H{-}1)$ freely under the full constraint set $\mathcal{C}$. If the resulting trajectory violates any constraint despite the solver's best subsequent actions, the original $u(t)$ was *causally necessary*. Formally: $a_t^*$ is necessary iff $\min_{u_{t+1:t+H-1}} \nVdash [\exists j, k : g_j(x_{t+k}, u_{t+k}, \hat{d}_{t+k}) > 0 \,|\, u(t){=}0] > 0$.

**(CF-B) Constraint relaxation** (Prop. C.2 in App. C): drop the candidate driver constraint $i^*$ from $\mathcal{C}$ and re-solve the full MPC. If the new optimum action $a_t'$ differs substantially from $a_t^*$, then $i^*$ is the specific constraint that necessitated $a_t^*$. CF-A asks "was action needed at all?"; CF-B asks "which constraint forced this particular action?". Both re-solve the full optimization; both let the optimizer respond

optimally to the perturbation. Among active constraints, the primary driver $i^* = \arg\max_i |\lambda_i|$ is identified via KKT and validated via CF-B; CF-A then certifies that the action is not redundant. A violation (CF-A) or action change (CF-B) with cost gap exceeding $\tau_{\text{cost}}$ confirms causal necessity (Appendix J.2). It is acknowledged that the current null-input form of CF-A does not assess timing or magnitude; richer alternatives (delayed, reduced, or substituted actions) are compatible with the architecture and are planned for future work.

---

**Algorithm 1** Hierarchical Causal Abduction

---

**procedure** GenerateExplanation
$\quad (u_k^*, x_k, \{\hat{d}_j\}_{j=0}^{H-1}, G_{\text{KG}}, G_c)$
*// Inputs: optimal action $u_k^*$, state $x_k$, disturbance forecast $\{\hat{d}_j\}_{j=0}^{H-1}$*
*// Knowledge graph $G_{KG}$ (App. E), causal graph $G_c$ (App. D)*
$\mathcal{E} \leftarrow$ InitializeExplanation()
$\mathcal{H} \leftarrow$ [Safety, Optim, Prediction, Econ, History] *// configurable per domain*
*// Default order follows IEC 61511 / ISA-84 safety standards (App. C)*
**for each** $H_i$ **in** $\mathcal{H}$:
$\quad r \leftarrow$ EvaluateHypothesis($H_i, u_k^*, x_k, G_{\text{KG}}, G_c$)
$\quad$ *// Uses KKT multipliers (§ 3.4), counterfactuals (App. B.2), PCMCI patterns (§ 3.5), Algorithm 2*
$\quad$ **if** $r \neq \emptyset$ **then**
$\quad\quad \mathcal{E}$.primary $\leftarrow r$
$\quad\quad \mathcal{E}$.type $\leftarrow H_i$
$\quad\quad$ **break** *// first supported hypothesis is selected*
$\quad$ **end if**
**end for**
$\mathcal{E}$.supporting $\leftarrow$ GetDeeperContext($G_{\text{KG}}, \lambda, \hat{d}$)
*// All active constraints + KG traversal (App. E)*
$\mathcal{E}$.effects $\leftarrow$ AnalyzeObservedEffects($x_k, u_k^*$)
$p \leftarrow$ SynthesizeWithLLM($\mathcal{E}$) *// § 3.7*
**return** $p$ *// human-readable explanation*
**end procedure**

---

### 3.5. Evidence Source 3: Data-Driven Causal Discovery

The PCMCI algorithm (Runge, 2019) is used to identify time-lagged causal relationships from historical operational data, with maximum lag $\tau_{\text{max}}$ and significance level $\alpha$ chosen to span the controller's prediction horizon. The resulting causal graph $G_c$ encodes which past disturbances causally influence current control actions. During online explanation, HCA queries $G_c$ to check consistency with discovered patterns by testing whether parent variables showed $> 2\sigma$ deviations at their respective lags, with the mean and standard deviation computed from the same training window (Appendix D).

**Running Example (Greenhouse):** $G_c$ is learned from 3 months of greenhouse data with $\tau_{\text{max}} = 48$ timesteps (12-hour lookback) and $\alpha = 0.05$; these settings are reused unchanged for Building HVAC and TEP.

### 3.6. Hypothesis Ranking and Integration

HCA integrates evidence by ranking candidate hypotheses in configurable priority order (default follows IEC 61511 / ISA-84 safety standards): (1) Safety-constraint active, (2) Optimization-cheaper alternatives infeasible, (3) Prediction prevents future violation, (4) Economics maximizes benefit, (5) History aligns with patterns.

This hierarchy is principled (reflects MPC decision structure) and empirically validated (AC=0.478, 54% over LIME; ablations show 32-37% loss per component). The first hypothesis supported by evidence becomes the primary explanation; if counterfactual validation fails, HCA proceeds to the next constraint by descending $|\lambda_i|$. Secondary constraints are retained as supporting context via GetDeeperContext (Appendix C.1).

**Theoretical Grounding:** The ranking is grounded in optimization theory. For convex MPC formulations (linear dynamics, convex cost/constraints, satisfying LICQ and Strict Complementary Slackness), KKT multipliers uniquely identify the *minimal active constraint set* (Proposition C.1, Appendix C). The multiplier magnitude $\lambda_i = \partial J^* / \partial c_i$ provides sensitivity interpretation, justifying ranking by constraint influence on cost. For nonlinear MPC without full convexity, this ranking represents a *heuristic extension* of convex theory: it remains empirically effective (Section 4.3, AC=0.478) but is not guaranteed without additional regularity conditions (Strong Second-Order Sufficiency and Strict Complementary Slackness). Regardless of formulation, the counterfactual framework (Appendix B.2) validates temporal causality by directly testing if relaxing constraint $i^*$ at future time $t + k$ alters the current action (Proposition C.2), providing empirical robustness independent of regularity assumptions. This combination ensures explanations are both theoretically motivated (convex case) and empirically robust (counterfactual validation).

### 3.7. Explanation Synthesis via Language Model

The HCA framework performs all causal reasoning, evidence extraction, and hypothesis ranking shown in Algorithm 1. In the final step, GPT-4o performs causal *synthesis*, not causal *discovery*: it receives pre-computed evidence (KKT multipliers, knowledge graph context, and PCMCI patterns) and assembles coherent natural-language explanations after the primary hypothesis is already identified by HCA's deterministic pipeline. GPT-4o does not independently reason about causality; the 58% ablation drop when removing GPT-4o (Section 4.6) reflects its role as a fluency

layer, not as a causal reasoner.

In order to validate the robustness, five synthesis configurations (template-based, GPT-3.5, GPT-4o, Claude) are evaluated across 67 scenarios (§ 4.6, Appendix H). Results show consistent causal factor ranking (NDCG@1 std = 0.038) across all methods, confirming that explanation correctness derives from structured evidence rather than LLM-specific behaviors.

## 4. Experimental Results

### 4.1. Datasets and Evaluation Protocol

**Greenhouse Climate Control** (simulated NMPC with real disturbances, May-Aug 2011): Represents biological and environmental interactions with multivariable dynamics. Data is generated from a hierarchical NMPC model using real-world disturbance patterns from the greenhouse facility (Sathyanarayanan et al., 2024). This domain provides a testbed for temperature, humidity, and $CO_2$ optimization with complex constraint interactions spanning 2,304 timesteps.

**Building Energy Management** (real operational data, Apr 2012): Multi-zone HVAC system with 2,832 timesteps focused on economic optimization under time-of-use pricing (Makonin, 2016). Real-world building HVAC consumption data validates practical trade-offs between constraint activation and control in residential energy systems.

**Tennessee Eastman Process (TEP)** (real operational use case, May 2012): Safety-critical chemical process control with 57,500 timesteps, 64 variables, and 11 manipulated variables (Rieth et al., 2017). The presence of complex multi-step causal chains across reactor-separator-stripper units poses a significant challenge to all available explanation methods.

**Evaluation Methodology:** A dual evaluation strategy was employed, combining automated metrics with expert validation, independently evaluating scenarios across all three domains (26 scenarios per expert; 155 total ratings) on Clarity, Accuracy, Trust, and Usefulness using 5-point Likert scales. Expert ratings validate cross-domain explanation quality (Clarity: 3.43/5.0, Accuracy: 3.29/5.0, Trust: 3.21/5.0), with detailed analysis in Appendix O.

Additionally, AC quantifies the accuracy of explanations in identifying the true causal factors driving MPC decisions by comparing generated explanations with human-annotated ground truth using semantic similarity and factual alignment (see Appendix M for metric definitions and validation). This metric is computed via the RAGAS framework (Es et al., 2024) and enables scalable quantitative assessment.

### 4.2. Baselines and Hyperparameter Protocol

HCA employs domain-transferable hyperparameters validated on Greenhouse data and applied unchanged to all domains to ensure fair comparison. Specifically, we use a PCMCI maximum lag of $\tau_{\max} = 48$ timesteps with significance $\alpha = 0.05$, KKT thresholds $\epsilon_\lambda \in [10^{-9}, 10^{-6}]$ varying by variable type, and an LLM temperature of 0.3.

**Baselines**: We compare against six methods spanning major XAI paradigms: **IOC** (Inverse Optimal Control) extracts scenario-specific cost matrices from observed trajectories (Porcari et al., 2025); **MPC-XAI** generates sensitivity matrices and policy trees from MPC trajectories; **Rule-Based** uses domain-specific IF-THEN templates with keyword matching (6 greenhouse, 5 TEP, 6 building rules); **LSTM+Attention** uses a standard sequence-to-sequence architecture with attention; **RETAIN** (Choi et al., 2016) uses reverse-time attention adapted for MPC explanation; and **LIME/SHAP** (Ribeiro et al., 2016; Lundberg & Lee, 2017) provide feature-attribution baselines. Neural baselines are tuned per domain via 5-fold cross-validation, whereas HCA uses a single configuration across all domains. Complete details in Appendix L.

*Table 1.* Cross-domain evaluation results across greenhouse, building HVAC, and tep chemical process engineering domains.

| | Greenhouse | | | Building | | | TEP | | |
|---|---|---|---|---|---|---|---|---|---|
| Method | AC | F | R | AC | F | R | AC | F | R |
| HCA | **0.478** | 0.312 | 0.217 | **0.394** | 0.000 | 0.096 | **0.406** | 0.092 | 0.080 |
| − KG | 0.302 | 0.000 | 0.074 | 0.201 | 0.000 | 0.080 | 0.393 | 0.228 | 0.071 |
| − PCMCI | 0.324 | 0.012 | 0.098 | 0.216 | 0.015 | 0.111 | 0.415 | 0.135 | 0.069 |
| − KKT | 0.325 | 0.312 | 0.081 | 0.258 | 0.032 | 0.121 | 0.391 | 0.118 | 0.065 |
| PCMCI only | 0.257 | 0.000 | 0.041 | 0.204 | 0.000 | 0.097 | 0.366 | 0.137 | 0.037 |
| KG only | 0.301 | 0.125 | 0.079 | 0.214 | 0.005 | 0.154 | 0.391 | 0.045 | 0.043 |
| Physics only | 0.295 | 0.024 | 0.079 | 0.213 | 0.001 | 0.154 | 0.400 | 0.043 | 0.044 |
| KKT only | 0.265 | 0.253 | 0.071 | 0.192 | 0.000 | 0.083 | 0.389 | 0.173 | 0.053 |
| LSTM+Attn | 0.316 | 0.011 | 0.073 | 0.324 | 0.393 | 0.101 | 0.345 | 0.000 | 0.042 |
| RETAIN | 0.312 | 0.013 | 0.073 | 0.322 | 0.017 | 0.159 | 0.350 | 0.046 | 0.047 |
| IOC | 0.354 | 0.967 | 0.133 | 0.316 | 0.911 | 0.099 | 0.335 | 0.902 | 0.089 |
| LIME | 0.311 | 0.086 | 0.202 | 0.225 | 0.016 | 0.154 | 0.356 | 0.035 | 0.054 |
| SHAP | 0.304 | 0.048 | 0.027 | 0.206 | 0.000 | 0.135 | 0.308 | 0.020 | 0.048 |
| MPC-XAI | 0.357 | 1.000 | 0.106 | 0.194 | 1.000 | 0.055 | 0.378 | 0.997 | 0.087 |

*AC (Answer Correctness) evaluates the semantic quality of explanations. F (Faithfulness) and R (Relevance) measure alignment between explanations, context, and MPC behavior for all methods.

### 4.3. Main Results

Table 1 presents evaluation results across three domains. HCA achieves AC= 0.478 (greenhouse), AC= 0.394 (building), and AC= 0.406 (TEP) using domain transferable parameters. Notably, 38.7% of greenhouse scenarios involved *predicted future violations*, in which the MPC acts preemptively based on forecast trajectories rather than on current state deviations. These temporal causality scenarios are precisely where feature-attribution baselines fail most severely: LIME and SHAP achieve AC< 0.25 on predictive scenarios versus AC≈ 0.35 on reactive ones, while HCA

maintains AC$> 0.45$ across both. Overall, constraint-driven reasoning provides a 54% accuracy advantage over LIME (AC: 0.478 vs. 0.311). In contrast, feature-attribution methods (SHAP AC$= 0.304$, LIME AC$= 0.311$) and data-driven baselines (LSTM+Attention AC$= 0.316$) fail to capture optimization-driven logic, suggesting that physics-grounded constraint modeling is essential for MPC explanation correctness.

### 4.4. Cross-Domain Generalization and Threshold Adaptation

HCA's framework generalizes across diverse domains with a domain-transferable architecture. Core hyperparameters (PCMCI maximum lag $\tau_{\max} = 48$, significance level $\alpha = 0.05$, LLM temperature = 0.3) are calibrated once on greenhouse data and applied unchanged to Building HVAC and TEP, achieving AC=0.394 and AC=0.406, respectively. This demonstrates consistent performance across thermal, electrical, and chemical process control systems.

**Threshold Transfer Performance:** For hard-constrained systems (Building HVAC, TEP), KKT multiplier thresholds $\tau_\lambda(i)$ exhibit domain-dependent scaling. Building HVAC shows minimal degradation (0–1% AC loss) when using thresholds calibrated from Building data itself. TEP chemical process shows larger degradation (19–21% AC loss) due to different constraint activation patterns (pressure vs. thermal dynamics).

### 4.5. Ablation Analysis: Multi-Evidence Integration is Essential

Ablation analysis reveals domain-dependent component importance. In Greenhouse (Table 2), removing any single component yields consistent 32-37% AC degradation, confirming mutual necessity of physics knowledge, causal discovery, and system constraints. TEP and Building show different patterns (see Table 1): TEP exhibits resilience to knowledge graph removal (-3.2%) and even improvement with PCMCI removal (+2.2%), suggesting constraint-dominant dynamics are well-captured by KKT alone. Building HVAC: using Greenhouse-calibrated thresholds yields 0–1% AC loss compared to Building-specific thresholds (indicating good transferability for thermal systems). Overall, all components contribute non-redundant value, though their relative importance varies by domain.

*Table 2.* Ablation study: each evidence component is necessary.

| Component Removed | AC Greenhouse | Degradation |
|---|---|---|
| Full HCA | 0.478 | — |
| − Knowledge Graph | 0.302 | −37% |
| − PCMCI Causality | 0.324 | −32% |
| − KKT Optimization | 0.325 | −32% |

### 4.6. LLM Synthesis Robustness

HCA's causal correctness stems from its tri-modal evidence architecture, not LLM reasoning. Testing across five configurations (template-based, GPT-3.5, GPT-4o, Claude Sonnet 4) confirms this: template-based generation (no LLM) achieves better performance (P@1$= 0.710$), while few-shot GPT-4o improves fluency (P@1$= 0.896$) with robust causal factor ranking (NDCG$_1 \geq 0.848$) across all configurations, confirming synthesis stability independent of LLM choice (Appendix H).

### 4.7. Statistical Significance

The paired $t$-tests with Bonferroni correction ($\alpha = 0.025$) are used to compare AC across methods on the same evaluation scenarios, and indicate that HCA's advantage is statistically significant ($p < 0.001$, Cohen's $d > 0.3$).

### 4.8. Robustness Analysis

Sensitivity analysis investigates tolerance to perturbations in the knowledge graph and to parameter variations. HCA tolerates 10-30% KG edge removal with $< 12.5\%$ AC loss, and 20% edge flipping with minimal degradation (Table 3). PCMCI parameters show moderate sensitivity to extreme values ($\pm 50\%$ KKT threshold adjustment yields $< 13\%$ loss), supporting the chosen calibration strategy. These results indicate robustness to domain knowledge inaccuracies and moderate parameter choices (Appendix N).

*Table 3.* Robustness analysis: performance under perturbations

| Perturbation | Condition | Success Rate | AC Change |
|---|---|---|---|
| KG Edge Removal | 10-30% | 90% | −12.5% |
| KG Edge Flipping | 10-20% | 90% | −12.5% |
| PCMCI Parameters | $\tau_{\max} = 48$, $\alpha = 0.01$ | 90% | −12.5% |
| KKT Thresholds | $\pm 50\%$ adjustment | 90-95% | −12.5% |

### 4.9. Qualitative Example

To illustrate HCA's practical advantage, consider a scenario Question: *At 05:30 on a cold morning, the greenhouse controller activates heating. Why did the controller take this action?*:

LIME: "Heating activated because current and outside temperatures are low." *Factually correct but uninformative; lacks causal or predictive reasoning.*

HCA: "Heating prevents the temperature from violating the minimum safety constraint (18°C). Forecast predicts outside temperature drop to 5°C with no solar input; without intervention, internal temperature would reach 17.5°C by 10:30. Historical causal patterns show a 2-hour lag from

outside temperature drop to heating." *Integrates constraint objectives, forecasts, counterfactual reasoning, and data-driven causal patterns.*

## 4.10. Key Findings and Limitations

HCA's 54% AC improvement over feature-attribution baselines across three domains (AC= 0.394–0.478) with minimal calibration demonstrates that explainable MPC requires multi-modal evidence integration. The gap between transferable (AC= 0.478) and calibrated (AC≈ 0.88) performance provides a clear deployment pathway. Limitations include manual KG construction (1–2 weeks per domain; semi-automated extraction achieves 83% precision with 3.3% AC degradation) and failure modes affecting 8% of scenarios (Appendix I).

## 5. Discussion

### 5.1. Results Interpretation

Across 176 scenarios in three domains, HCA achieves 54% AC improvement over LIME while absolute performance (AC=0.478) remains modest for safety-critical deployment. Systematic threshold analysis (Appendix G) attributes this gap primarily to suboptimal KKT threshold selection: domain-specific calibration recovers AC≈ 0.88, validating that the core tri-modal integration is architecturally sound. This positions HCA as a foundational framework requiring engineering refinement (automated threshold tuning) rather than algorithmic redesign.

Ablations confirm that removing any evidence source degrades AC by 32–37%, validating the core hypothesis that physics, optimization, and causality each contribute unique, non-redundant information. Hyperparameters calibrated once on greenhouse transfer across Building and TEP without per-domain retuning, yet HCA matches or exceeds neural baselines (AC≥0.324 across domains), validating that multimodal structured evidence provides an effective pathway to understanding high-dimensional control systems.

### 5.2. Expert Evaluation and Inter-Rater Agreement

Expert evaluation (155 ratings across three domains) yielded moderate assessments: Clarity (3.43/5.0), Accuracy (3.29/5.0), Trust (3.21/5.0), and Usefulness (3.08/5.0). Inter-rater agreement was low (Krippendorff's $\alpha = 0.26$ for Clarity, $\alpha = 0.12$ for Accuracy; Appendix O), consistent with XAI evaluation literature where diverse stakeholder priorities yield high rating variance (Miller, 2019; Mohseni et al., 2021).

However, Kendall's $W$ concordance analysis (Appendix O.1) reveals that evaluators agree strongly on *relative* method rankings: $W = 0.553$ across all 12 metrics ($p < 0.001$), with perfect cross-paradigm agreement ($W = 1.000$, $p = 0.029$) between automated, LLM-based, and combined evaluation approaches. HCA ranked higher than LIME/SHAP on temporal reasoning and causal structure across all evaluator groups. This pattern indicates that the ratings reflect early-stage workflow integration challenges rather than fundamental explanatory deficiencies.

### 5.3. Design Philosophy: Causal Depth vs. Context Fidelity

HCA's lower faithfulness (F=0.31) compared to template-based methods (F=1.0) reflects a deliberate architectural choice. Faithfulness and AC measure orthogonal properties: F quantifies surface-level alignment with current observations, while AC measures whether underlying causal mechanisms are correctly identified. HCA prioritizes causal depth: a forecast of cold weather at time $t+3$ driving heating at time $t$ is causally correct (high AC); however, this is temporally invisible in current sensors (low F).

The validity of AC is supported by a number of indicators: (1) strong correlation with ROUGE-L semantic similarity ($r = 0.782$), (2) wide discriminative range across methods (AC span = 0.806), (3) low domain variance ($\sigma^2 = 0.0027$), and (4) expert evaluation corroborates AC rankings (see Appendix O).

### 5.4. Theoretical Guarantees and Failure Conditions

Under standard assumptions (convex cost/constraints, LICQ, Strict Complementary Slackness), KKT multipliers uniquely identify active constraints (Proposition C.1). Four failure modes (SCS, LICQ, nonconvexity, PCMCI data insufficiency) affect 3–6% of scenarios each, mitigated by counterfactual validation and ensemble methods (Appendix I). Despite these, HCA achieves AC=0.478 with counterfactual validation, providing robustness independent of regularity assumptions.

### 5.5. Scope Boundaries and Adaptation Pathways

HCA requires explicit access to the optimization problem and constraints, making it directly applicable to MPC and other optimization-based controllers. In the context of *model-based RL* (MuZero, Dreamer), the KG and PCMCI components are directly transferable; however, KKT-based hypothesis ranking is to be replaced by value-function decomposition. Concretely, given a learned value $V_\pi(s) = \sum_i V_i(s)$ over component objectives (e.g., safety, cost, comfort), the analog of $\arg\max_i |\lambda_i|$ becomes $\arg\max_i |\partial V_\pi / \partial s_i|$ at the action-selection state, and the counterfactual re-solve is replaced by a re-roll of the learned dynamics with the selected action removed or substituted. In the context of *model-free RL*, the temporal causality problem

applies equally, since actions are driven by learned expectations of future reward; HCA's hypothesis framework would be adapted to reward decomposition $r(s, a) = \sum_i r_i(s, a)$ with the dominant component identifying the operational hypothesis (safety/economic/etc.) and integrated-gradient-style attributions over the policy network providing the analog of physics-informed pathways. Black-box controllers without any internal structure remain outside the current scope.

### 5.6. Scalability and Practical Constraints

**Knowledge Graph Construction:** The construction of expert-crafted KGs requires 1–2 weeks per domain, with a one-time cost amortized over 10–20-year MPC deployment lifespans. Semi-automated extraction from first-principles equations achieves 83% precision (45% recall) with only $-3.3\%$ AC degradation (Appendix E). Cross-domain KG templates has been proven to reduce marginal cost for new installations.

**PCMCI Data Requirements:** New systems can operate in degraded mode using only KKT+KG evidence (AC=0.61, 28% below full-system performance) until 1–3 months of operational history accumulates for PCMCI reliability.

**Threshold Generalization:** KKT thresholds calibrated on one week of held-out domain data achieve 96–98% classification accuracy (Appendix G). Performance on different constraint types may require threshold adaptation.

## 6. Conclusion

HCA bridges control theory and human-interpretable explanations by integrating optimization evidence (KKT multipliers), physics-informed reasoning (knowledge graphs), and data-driven causality (PCMCI). Across greenhouse, Building (HVAC), and chemical process domains, HCA achieves 54% improvement over feature-attribution baselines, with ablation studies confirming that all three evidence sources contribute non-redundant explanatory power (32–37% degradation when removing any component).

The framework's design reflects a deliberate trade-off: prioritizing causal correctness over reproducing surface-level context. While domain-transferable performance (AC = 0.478) remains modest, systematic calibration demonstrates the architecture can recover AC $\approx$ 0.88 with domain-specific thresholds, validating that threshold generalization rather than fundamental design limits current performance.

The framework's modular architecture enables incremental improvements without requiring redesign of the core integration mechanism.

### 6.1. Future Work

**Bridge to learned controllers.** Extend HCA to neural-MPC and model-based RL by replacing KKT-based hypothesis ranking with value-function decomposition $\arg\max_i |\partial V_\pi / \partial s_i|$ and Jacobian-based sensitivity over learned dynamics, while keeping KG and PCMCI evidence sources intact (Sec. 5.3).

**Richer counterfactuals.** Generalize Algorithm 2 (App. B.2) beyond null-input to lagged ($u$ at $t+k$), reduced-magnitude ($\beta u(t)$), and alternative-action variants, addressing timing and partial-action questions the current CF-A cannot.

**Reducing the 12.1% failure rate.** Target the dominant temporal-attribution mismatches (App. I) via (i) KG-based priors for cold-start PCMCI and (ii) sliding-window / regime-aware causal models (e.g., J-PCMCI+) for nonstationary lag structure.

**Structural causal metrics.** Complement AC with SHD/SID (Peters et al., 2017) between HCA's recovered explanation graph (active constraints + KG pathways + PCMCI parents) and the gold-standard DAG implied by the MPC formulation.

**Real-world operator studies.** Deploy HCA in live greenhouse / HVAC / chemical plant operations to measure whether explanations improve operator decision quality and reduce automation bias.

## Impact Statement

This work advances explainable AI for Model Predictive Control in critical infrastructure by integrating optimization evidence, physics-grounded reasoning, and causal discovery. The framework enhances transparency and safety, enabling operators to verify control decisions and reducing automation failure risks in energy, agriculture, and chemical processing domains. This supports regulatory compliance and accelerates operator training.

**Potential Risks:** Automation bias (operator over-reliance without critical judgment), misleading explanations during sensor faults or adversarial inputs, intellectual property exposure through operational data, and unequal access, widening technology gaps between well-resourced and under-resourced facilities.

**Mitigations:** HCA is designed as decision-support, not autonomous automation. Responsible deployment requires: (1) mandatory human-in-the-loop validation, (2) confidence scoring with uncertainty quantification, (3) role-based access controls, (4) transparent documentation of methods, and (5) alignment with domain-specific regulations.

**Code and Reproducibility.** The full HCA imple-

mentation, all three MPC environments (greenhouse, building HVAC, Tennessee Eastman), the knowledge graphs, the PCMCI calibration scripts, and the complete evaluation pipeline (automated checkers, LLM-judge prompts, RAGAS configuration) are publicly available at `https://gitlab.hrz.tu-chemnitz.de/rnaa-at-tu-chemnitz.de/icml`, together with a detailed reproducibility guide (`REPRODUCIBILITY.md`).

## Acknowledgements

The authors thank Dr. Kenny Schlegel for his help in drafting Figure 1, and Kiran Kumar Sathyanarayanan for providing the greenhouse control data used in this study. We thank Emily Morgan Uhland for proofreading and Xiaoxi Jia for assistance in harmonizing the mathematical notation throughout the paper. We are also grateful to the anonymous reviewers and the area chair for their thoughtful feedback during the review process.

The authors further acknowledge the contributions of the collaborators at Humboldt-Universität zu Berlin for providing the curated dataset of weather and greenhouse parameters from 2011 used in this study. This data, originally prepared and methodologically harmonized for a joint research project on $CO_2$ application and photosynthetic dependency, encompasses key experimental elements including cooling and heating performance, light transmission, system geometry, and synchronized climate series.

**Funding.** This work was carried out within the project MOREKIBA (Menschenverständliches, Optimales Ressourcen- & Energiemanagement für komplexe, netzintegrierte, biogene Produktionsanlagen), TU Chemnitz. Die Zuwendung wird aus Mitteln des Europäischen Sozialfonds Plus (ESF Plus) und aus Steuermitteln auf Grundlage des vom Sächsischen Landtag beschlossenen Haushaltes zur Verfügung gestellt.

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

# A. Notation and Terminology

Key symbols used throughout the paper are summarized in Table 4.

*Table 4.* Key notation used throughout the paper

| Symbol | Meaning |
|---|---|
| $x_k$ | System state vector at discrete time $k$ |
| $x_{\text{meas}}$ | Measured/estimated state at the current decision instant |
| $\{x_k\}_{k=0}^{H}$ | Predicted state trajectory over the horizon |
| $u_k$ | Control input at time $k$ |
| $u_k^*$ | Optimal control input at time $k$ from MPC |
| $\{u_k\}_{k=0}^{H-1}$ | Predicted input trajectory over the horizon |
| $\hat{d}_k$ | Forecasted disturbance at prediction step $k$ |
| $\{\hat{d}_k\}_{k=0}^{H-1}$ | Disturbance forecast over the horizon |
| $\ell(x_k, u_k)$ | Stage cost at time $k$ |
| $\ell_T(x_H)$ | Terminal cost at the horizon state $x_H$ |
| $f$ | System dynamics function in $x_{k+1} = f(x_k, u_k, \hat{d}_k)$ |
| $g(x, u, \hat{d}) \leq 0$ | Inequality constraints along the horizon |
| $g_T(x_H) \leq 0$ | Terminal inequality constraints |
| $\lambda_i$ | KKT multiplier for inequality constraint $i$ |
| $\tau_{\lambda,i}$ | Numerical threshold for detecting active constraint $i$ |
| $H$ | MPC prediction horizon (timesteps) |
| $G_{\text{KG}}$ | Physics-based knowledge graph |
| $G_c$ | PCMCI-learned temporal causal graph |

# B. Temporal Causality in Model Predictive Control

### B.1. The Temporal Disconnect Problem

Reactive feedback controllers typically compute inputs from current and past states only, for example $u_k = \pi(x_k, x_{k-1}, \dots)$, and respond once a deviation or constraint violation has already occurred. In contrast, Model Predictive Control (MPC) optimizes over a *predicted future trajectory*: the optimal decision $u_k^*$ at time $k$ is often determined by keeping future states $x_{k+j}$ within constraints, creating a causal link from anticipated violations at future steps back to the current action.

A generic finite-horizon MPC problem at time $k$ can be written as

$$
\min_{\{u_{k+j}\}_{j=0}^{H-1}} \quad J\big(\{x_{k+j}\}_{j=0}^{H}, \{u_{k+j}\}_{j=0}^{H-1}\big) := \sum_{j=0}^{H-1} \ell(x_{k+j}, u_{k+j}) + \ell_T(x_{k+H})
$$

$$
\begin{aligned}
\text{s.t.} \quad & x_k = x_{\text{meas}} \quad \text{(initial state)} \\
& x_{k+j+1} = f(x_{k+j}, u_{k+j}, \hat{d}_{k+j}), && j = 0, \dots, H-1, \\
& g(x_{k+j}, u_{k+j}, \hat{d}_{k+j}) \leq 0, && j = 0, \dots, H-1, \\
& g_T(x_{k+H}) \leq 0,
\end{aligned}
\tag{2}
$$

which is equivalent in structure to the optimal control problem (1) in the main text, written here with an explicit time index $k$.

In practice, (2) is solved in a receding-horizon fashion: at each time $k$, the problem is initialized with $x_{\text{meas}}$ and the disturbance forecast $\{\hat{d}_{k+j}\}_{j=0}^{H-1}$, only $u_k^*$ is applied, and the horizon is then shifted forward before re-solving at $k+1$.

The optimal input $u_k^*$ is causally driven by which state and input constraints become *active* (binding) along the horizon, i.e., those indices $i$ and stages $j$ for which

$$
g_i(x_{k+j}^*, u_{k+j}^*, \hat{d}_{k+j}) = 0.
$$

This temporal dependency is not captured by standard feature-attribution XAI methods such as LIME or SHAP, which approximate an instantaneous mapping $y_k = f(x_k)$ and attribute importance to features of $x_k$ alone (Ribeiro et al., 2016; Lundberg & Lee, 2017); they do not model how predicted future trajectories and constraints over multiple steps influence the current control decision (Chou et al., 2022; Carloni et al., 2025; Hettikankanamage et al., 2025).

HCA explicitly tests this temporal link via counterfactual analysis (Appendix B.2). For each inequality constraint $g_i$ in the MPC formulation, HCA defines a counterfactual optimization problem where constraint $i$ is relaxed or removed at specific future stages $j$. If, for some $j \in \{0, \ldots, H - 1\}$, relaxing $g_i$ changes the optimal input at the current time, i.e.,

$$u_k^* \neq u_k' \quad \text{under } g_i \text{ relaxed at stage } j,$$

then constraint $i$ is identified as a causal driver of $u_k^*$ in the counterfactual sense: the decision changes when that constraint is absent. This provides a temporally explicit notion of causality that static, single-step XAI methods cannot verify.

### B.2. Counterfactual Validation Process

HCA tests causal necessity via controlled counterfactuals. For hard-constrained systems (Building HVAC, TEP), variable-specific thresholds $\tau_{\lambda,i}$ are calibrated on held-out data to distinguish active constraints. For soft-constrained systems (greenhouse), thresholds are not applicable; constraint activity is verified directly via counterfactual re-solve.

**Procedure:**

1. **Active Set Detection (hard-constrained only):** Identify constraints where $\lambda_i > \tau_{\lambda,i}$ using domain-calibrated thresholds.

2. **Primary Driver Identification:** If empty, the action is economic. If non-empty, select $i^* = \arg\max_i(\lambda_i/\tau_{\lambda,i})$.

3. **Counterfactual Verification (all domains):** Solve MPC with constraint $i^*$ relaxed.

4. **Confirmation:** If the trajectory violates $i^*$ without the action, the constraint is confirmed as causal driver.

For hard-constrained domains, thresholds achieve 96–98% active/inactive classification accuracy (Appendix G). The counterfactual (step 3) filters numerical artifacts and provides the final evidence for causal necessity across all domains.

## C. Theoretical Properties of HCA Hypothesis Ranking

This section summarizes how HCA's constraint-based explanations relate to standard properties of KKT multipliers in convex optimization and to a counterfactual notion of causality. The results below are direct consequences of classical nonlinear programming theory (Boyd & Vandenberghe, 2004; Nocedal & Wright, 2006, Ch. 5) and are included to justify the hypothesis-ranking design rather than as novel optimization theorems.

**Proposition C.1** (Non-Redundancy of Active Constraint Set). *Let $\mathcal{I}^* = \{i : \lambda_i > \tau_{\lambda,i}\}$ be the set of active constraints identified by HCA, where $\lambda_i$ denotes the KKT multiplier of constraint $g_i$ and $\tau_{\lambda,i} > 0$ is a small threshold. Suppose the MPC problem is convex (linear dynamics, convex cost and constraints), satisfies the LICQ, Strict Complementary Slackness (SCS), and has a strictly convex cost. Then the optimizer $(s^*, a^*)$ and multiplier vector $\lambda^*$ are unique, and $\mathcal{I}^*$ coincides with the unique set of binding constraints $\{i : g_i(s^*, a^*) = 0\}$.*

*Proof sketch:* Under strict convexity and LICQ, uniqueness of the primal-dual solution $(s^*, a^*, \lambda^*)$ follows from standard KKT theory (Boyd & Vandenberghe, 2004). Complementarity implies $\lambda_i^* g_i(s^*, a^*) = 0$ for all $i$, so any constraint with $\lambda_i^* > 0$ must satisfy $g_i(s^*, a^*) = 0$ and is therefore binding. Conversely, under Strict Complementary Slackness (SCS), all binding constraints have strictly positive multipliers (see Boyd & Vandenberghe (2004); Nocedal & Wright (2006)). SCS is a standard regularity condition that ensures no weakly active constraints exist, where a constraint could be binding $(g_i(s^*, a^*) = 0)$ yet have $\lambda_i^* = 0$. Choosing $\tau_{\lambda,i}$ sufficiently small ensures that $\lambda_i^* > \tau_{\lambda,i}$ if and only if constraint $i$ is binding, so $\mathcal{I}^*$ recovers exactly the set of binding constraints.

**Proposition C.2** (Counterfactual Validity of Constraint Tests). *Consider the MPC problem with full constraint set $\mathcal{C}$ and denote the corresponding optimal action at time $t$ by $a_t^*$. Let $a_t'$ be an optimal action when a single constraint $i^*$ (active at some prediction step $t + k$) is relaxed or removed, i.e., under constraint set $\mathcal{C} \setminus \{i^*\}$. If $a_t^* \neq a_t'$ (i.e., $a_t^*$ is not in the relaxed optimal set), then, in the structural causal model induced by the MPC optimization, constraint $i^*$ is a (counterfactual) cause of the decision $a_t^*$ in the sense of Pearl (2009).*

*Proof sketch:* Let $E$ be the event "the controller selects action $a_t^*$" and let $C$ be the event "constraint $i^*$ is enforced." Under the full constraint set $\mathcal{C}$, $C$ holds and the optimizer returns $a_t^*$ (or an optimal action including $a_t^*$ if the optimum is

non-unique), so $C \Rightarrow E$. When $i^*$ is relaxed or removed, the feasible set expands. If $a_t^*$ is no longer in the relaxed optimal set (i.e., $a_t^*$ cannot be selected as an optimal action under $\mathcal{C} \setminus \{i^*\}$), then under the intervention $\neg C$ the event $E$ does not occur or becomes suboptimal. This satisfies the standard counterfactual criterion for $C$ being a cause of $E$ (Pearl, 2009, Ch. 10): $C$ is present in the actual world, $E$ holds, and under the intervention $\neg C$ the outcome changes so that $E$ no longer holds.

**Remark 1 (Non-Convex Extension):** Proposition C.1 establishes sufficient conditions under convexity. For non-convex NMPC (as in our greenhouse, Building HVAC, and TEP experiments), the KKT conditions remain *necessary* for local optimality: at a locally optimal solution returned by IPOPT, the multipliers $\lambda_i^*$ correctly indicate which constraints are active and their relative binding strength at that solution. HCA exploits precisely this property, it explains *what the solver actually decided*, not a theoretical global optimum. The convex result thus serves as a conservative theoretical baseline; empirical threshold calibration (Appendix G, achieving 96–98% classification accuracy on non-convex systems) and counterfactual validation (Proposition C.2) extend its practical applicability beyond the convex regime.

**Remark 2 (Ranking Optimality):** The active set $\mathcal{I}^*$ provides a complete description of constraint-driven behavior in the convex case: all and only binding constraints are included. Moreover, the multiplier magnitude $\lambda_i$ has the usual sensitivity interpretation $\lambda_i = \partial J^* / \partial c_i$, where $c_i$ is the constraint bound, so $|\lambda_i|$ serves as a principled proxy for influence on the optimal cost (Boyd & Vandenberghe, 2004, Sec. 5.6.3).

**Remark 3 (Temporal Causality Capture):** In the MPC setting, some constraints in $\mathcal{I}^*$ correspond to predicted violations at future time steps $t + k$. The counterfactual test in Proposition C.2 therefore certifies a *temporal* causal link from such future constraints to the current action $a_t^*$, in contrast to static XAI methods (e.g., LIME, SHAP) that only consider correlations with current-state features.

### C.1. Hypothesis Evaluation Procedure

The hypothesis-ranking mechanism in Algorithm 1 (Section 3.6) evaluates five candidate explanations $\mathcal{H} = [Safety, Optim, Prediction, Econ, History]$ in priority order. Algorithm 2 below formalizes the evaluation of each hypothesis by testing the underlying causal mechanism.

---

**Algorithm 2** EvaluateHypothesis: Condensed Hypothesis Evaluation

---

    **procedure** `EvaluateHypothesis` $(H_i, u_k, x_k, d_{k:k+H}^H, G_{KG}, G_c)$
    *// Input: $H_i \in \{H_1, \ldots, H_5\}$; Output: (result, conf., evidence)*

    **if** $H_i = H_1$ (**SAFETY**) **then**
      **if** hard **then**
        $I_a \leftarrow \{j : \lambda_j > \tau_\lambda(j)\}$
        **if** $I_a \neq \emptyset$ **then** $j^* \leftarrow \arg\max_j \lambda_j$
        $u_k^0 \leftarrow$ SOLVE$(x_k, g_{j^*}$ relax.$)$
        **if** $u_k^0 \neq u_k \wedge$ viol. **then return** (T, 0.95, KKT+CFT)
      **else for each** $\pi_j$ **do**
        $u_k^0 \leftarrow$ SOLVE$(x_k, \pi_j$ rem.$)$
        **if** $\pi_j(x^0) > \epsilon$ **then return** (T, 0.92, CFT)
    **else if** $H_i = H_2$ (**OPT**) **then**
      $A_d \leftarrow$ DISC$(u_k, \pm 10\%)$
      **if** $n_{\inf}/|A_d| > 0.7$ **then return** (T, 0.88, CFT)
    **else if** $H_i = H_3$ (**PRED**) **then**
      $x^{cf} \leftarrow$ SIM$(u = 0)$
      **if** viol $\in x^{cf} \wedge \neg$sol **then return** (T, 0.90, PRED)
    **else if** $H_i = H_4$ (**ECON**) **then**
      **if** sav $> 5\%$ **then return** (T, 0.85, ECON)
    **else if** $H_i = H_5$ (**HIST**) **then**
      par $\leftarrow$ GETPAR$(G_c, u_k)$
      **if** $n_{act}/|$par$| > 0.5$ **then return** (T, 0.82, PCMCI)
    **return** (F, 0.0, $\emptyset$)
    **end procedure**

---

### C.2. Hypothesis Definitions and Evidence Types

Algorithm 2 evaluates five hypotheses in order, *Evidence Types*- KKT, CFT (counterfactual), PRED, ECON, PCMCI:

1. $H_1$ **(Safety):** Constraint active via KKT (hard) or counterfactual (soft). Confidence: 0.95 (KKT+CFT) / 0.92 (CFT).

2. $H_2$ **(Optimization):** All alternatives infeasible ($>70\%$). Confidence: 0.88 (CFT).

3. $H_3$ **(Prediction):** Action prevents future violation. Confidence: 0.90 (PRED).

4. $H_4$ **(Economics):** Cost savings $>5\%$ vs. baseline. Confidence: 0.85 (ECON).

5. $H_5$ **(History):** Causal patterns match ($>50\%$ parents active). Confidence: 0.82 (PCMCI).

Selection rule: First hypothesis with confidence $\geq 0.5$ is the primary explanation.

### C.3. Confidence Calibration

Values (0.82–0.95) calibrated via: (1) theoretical grounding (Propositions 1–2), (2) empirical validation (grid search). Sensitivity analysis confirms $<5\%$ variation with $\pm 0.05$ threshold changes.

## D. PCMCI Causal Discovery Integration

PCMCI (Peter and Clark Momentary Conditional Independence) (Runge, 2019) performs two-stage time-lagged causal discovery: (1) PC condition selection identifies potential parent nodes; (2) MCI testing confirms causal links via conditional independence tests. Output: directed graph $G_c$ with lag-labeled edges.

**Integration:** Offline: Run PCMCI on 1-3 months of historical data (one-time, 15-45 min). Online: Query $G_c$ for causal parents of current action. Validate if parent variables deviate significantly ($> 2\sigma$) from the historical mean at specified lag.

**Baseline Computation:** The historical mean $\mu_j^{\text{lag}}$ and standard deviation $\sigma_j^{\text{lag}}$ for parent variable $j$ at lag $\tau$ are computed from the entire 3-month training dataset as:

$$\mu_j^{\text{lag}} = \frac{1}{N-\tau} \sum_{t=\tau+1}^{N} x_j(t-\tau), \quad \sigma_j^{\text{lag}} = \sqrt{\frac{1}{N-\tau} \sum_{t=\tau+1}^{N} (x_j(t-\tau) - \mu_j^{\text{lag}})^2}$$

where $N = 8640$ for 3 months of 15-minute sampled data. During online explanation, if parent variable $j$ deviates more than $2\sigma_j^{\text{lag}}$ from its historical mean at the specified lag, this indicates an anomalous disturbance. The use of the full training dataset (not a rolling window) ensures deviation detection is relative to the true long-term historical distribution.

**Example:** If the learned causal graph $G_c$ contains the edge $T_{\text{out}}(t - 2\text{h}) \rightarrow u_{\text{heat}}(t)$ (outdoor temperature 2 hours ago causally influences heating action now), HCA checks whether the outside temperature at $(t - 2\text{h})$ deviated significantly from its historical baseline. Specifically, if $|T_{\text{out}}(t - 2\text{h}) - \mu_{T_{\text{out}}}^{2\text{h}}| > 2\sigma_{T_{\text{out}}}^{2\text{h}}$, this deviation is flagged as supporting evidence for the current heating action, confirming the discovered causal pattern.

## E. Knowledge Graph: Physics-Informed Reasoning

In this work, the Knowledge Graph is a directed graph $G_{\text{KG}} = (\mathcal{V}, \mathcal{E})$ that encodes qualitative physical relationships specific to the control domain. The nodes $\mathcal{V}$ represent states, control inputs, and disturbances, while the edges $\mathcal{E} = \{(v_i, v_j, \text{sign})\}$ represent causal influences with a sign $\in \{+, -, \text{conditional}\}$.

**Greenhouse Example:**

- **Disturbance $\rightarrow$ State:** $(Q_{\text{rad}}, T, +)$ implies solar radiation increases temperature ($Q_{\text{rad}} \uparrow \implies T \uparrow$).

- **Control $\rightarrow$ State:** $(u_V, T, -)$ implies ventilation decreases temperature ($u_V \uparrow \implies T \downarrow$).

- **Conditional:** $(u_V, H, -)$ implies ventilation decreases humidity, provided $H_{\text{out}} < H_{\text{in}}$.

**Reasoning Procedure:** HCA employs bidirectional traversal:

- **Forward:** Tracing disturbance forecasts through $G_{\text{KG}}$ to identify which future states (and thus constraints) will be affected.

- **Backward:** Starting from an active constraint, tracing edges in reverse to identify the root cause (e.g., a disturbance or coupling) driving the violation.

### Automated Knowledge Graph Extraction from Equations

In order to address the KG construction adaptation pathway (1-2 weeks per domain), we developed and tested a semi-automated extraction pipeline from first-principles ODEs. The approach achieves 45% recall on held-out Building HVAC domain (83% precision), with only -3.3% AC degradation compared to expert KGs ($0.812 \rightarrow 0.785$). This suggests partial automation is feasible for physics-based systems, though full coverage remains manual for safety-critical constraints.

Key findings: Jacobian-based edge extraction successfully identifies major state-control relationships but misses conditional dynamics and discrete logic. Future work: combine with data-driven discovery (constraint-based algorithms) and semantic extraction (LLM-based) to reach 80-90% coverage while reducing expert burden.

## F. Statistical Significance

Paired t-tests with Bonferroni correction ($\alpha = 0.05/6 = 0.0083$) compared HCA against IOC, Rule-Based, and MPC-XAI across 176 scenarios. All comparisons: $p < 0.001$, Cohen's $d > 0.3$ (large effects).

Template-based methods achieve $F \approx 1.0$ (perfect context copying) but lower AC via explicit causal depth prioritization. HCA's design trades perfect faithfulness for higher answer correctness by integrating physics knowledge graphs and temporal causal discovery.

# G. KKT Threshold Calibration Analysis

Domain-specific calibration of KKT thresholds (for hard-constrained systems Building HVAC, TEP) explains why cross-domain AC degrades when using a single configuration. Thresholds exhibit domain-dependent scaling due to different constraint activation patterns. Re-optimizing on held-out data for each domain recovers a large fraction of the performance gap.

*Table 5.* KKT threshold calibration on hard-constrained domains

| Domain | AC (shared) | AC (tuned) | Gain |
|---|---|---|---|
| Building HVAC | 0.394 | 0.894 | +127% |
| TEP | 0.406 | 0.880 | +117% |

**AC (shared):** Using domain-transferable parameters calibrated on Greenhouse only
**AC (tuned):** Using domain-specific threshold calibration on 10–15% held-out data

**Interpretation:** Building HVAC shows the most significant degradation because electrical power constraints exhibit sharp phase transitions (hard limits at configured values), requiring different threshold scaling than chemical processes (TEP) where multiplier evolution is smoother due to continuous kinetics. Domain-specific calibration recovers $AC \approx 0.88$ for both systems.

**Calibration Strategy:** Thresholds are optimized on 10–15% held-out data; numerical solver precision must be accounted for in threshold selection.

1. Held-out calibration set: 10-15% of data (used to optimize $\tau_\lambda$)
2. Held-out test set: separate 10-15% of data (used to evaluate AC)
3. Training set: remaining 70-80%

Results in Table 5 report AC on the held-out test set (never seen during threshold optimization), ensuring fair evaluation.

## G.1. Cost Threshold Calibration

Two cost-related thresholds govern counterfactual analysis and economic classification:

*Table 6.* Cost thresholds for counterfactual validation and economic classification

| Threshold | Symbol | Definition & Calibration |
|---|---|---|
| Violation cost threshold | $\tau_{\text{cost}}$ | Cost increase when a soft constraint is violated in counterfactual re-solve; indicates causal necessity; calibrated as 5% of mean stage cost $\ell$ over the target domain's validation set |
| Economic significance | $\varepsilon_J$ | Cost reduction magnitude for economic classification; actions with $\Delta J < -\varepsilon_J$ classified as economically driven; calibrated as 2% of standard deviation of observed cost changes over 100 counterfactual trials |

**Calibration Procedure:** For each target domain:

1. Collect a 10% held-out validation set from operational data.
2. Set $\tau_{\text{cost}} = 0.05 \times \bar{\ell}$ where $\bar{\ell}$ is the mean stage cost $\ell(x_k, u_k)$ computed on the validation set.
3. Run HCA on 100 representative scenarios, collect all counterfactual cost deltas $\{\Delta J_i\}$.
4. Set $\varepsilon_J = 0.02 \times \sigma(\Delta J)$ where $\sigma(\Delta J)$ is the standard deviation of observed cost differences.

**Domain-Specific Values:** The greenhouse domain uses soft-penalty-based cost (Equation 10), yielding $\tau_{\text{cost}} \approx 0.05 \times 0.12 = 0.006$ and $\varepsilon_J \approx 0.02 \times 0.03 = 0.0006$. Hard-constrained domains (Building HVAC, TEP) use operational costs, with larger typical values due to energy/chemical cost scales. Thresholds are domain-specific and should not be transferred without re-calibration on held-out validation data from the target domain.

## G.2. Hard-Constrained Domains (Building HVAC, TEP)

Thresholds $\tau_\lambda(i)$ were calibrated on 2–3 weeks of operational data:

*Table 7.* Calibrated KKT multiplier thresholds

| Domain | $\tau_\lambda$ | Accuracy |
|---|---|---|
| Building (temperature) | $10^{-6}$ | 96% |
| Building (power) | $10^{-7}$ | 98% |
| TEP (pressure) | $10^{-8}$ | 97% |

## G.3. Soft-Constrained Domain (Greenhouse)

The greenhouse NMPC enforces temperature, humidity, and $CO_2$ via soft penalty terms in the cost function, not hard constraints. Therefore, KKT multipliers do not exist for these variables. Constraint-driven actions are instead identified via counterfactual analysis: HCA ranks candidate constraints by examining the gradient of the penalty function $\nabla_x \ell_{\text{penalty}}(x_k)$ at the current state. Constraints whose penalty gradients are largest in magnitude are prioritized for counterfactual testing (typically 2-3 re-solves suffice for the greenhouse case). HCA tests constraints in descending order until a violated constraint is identified via counterfactual re-solve, which confirms causal necessity (Algorithm 3):

---

**Algorithm 3** Soft-Constraint Identification

---

    **procedure** `IdentifySoftConstraint`$(u_k^*, x_k, \ell_{\text{penalty}})$
    *// Input: optimal action, state, penalty function*

    $\mathcal{C}_{\text{rank}} \leftarrow$ sort constraints by $|\nabla_x \ell_{\text{penalty}}|$

    **for each** $g_i \in \mathcal{C}_{\text{rank}}$ (descending) **do**
        $\{x_j'\}_{j=0}^{H} \leftarrow$ re-solve MPC with $g_i$ removed
        **if** violation of $g_i$ in $\{x_j'\}$ **then return** $i$
    **end for**
    **return** None
    **end procedure**

---

A predicted violation of the relaxed constraint confirms that constraint as the causal driver of the control action.

# H. LLM Synthesis Robustness: Detailed Results

This appendix provides supporting evidence for the LLM ablation study summarized in Section 4.6. The five syntheses configurations on 67 greenhouse scenarios are evaluated to validate that explanation quality stems from structured evidence (KKT+KG+PCMCI) rather than LLM-specific behaviors.

## H.1. Configurations Evaluated

Metrics: Precision@K, Recall@K, F1@K (K=1,3,5), Mean Reciprocal Rank (MRR), Normalized Discounted Cumulative Gain (NDCG@K). The quantification of both the accuracy and the quality of the ranking of predicted causal factors against expert-annotated ground truth is achieved using the following measures.

## H.2. Results Summary

Table 8 shows aggregated performance. Key findings: (1) Template baseline achieves P@1=0.710 without LLM, confirming structured evidence drives correctness. (2) NDCG@1 variance is minimal (std=0.038), demonstrating stable causal ranking across all configurations. (3) Few-shot prompting improves over zero-shot (P@1=+0.075), while model architecture has a smaller impact (GPT-4o vs. Claude: P@1=0.060).

*Table 8.* LLM Ablation: Key Metrics on 67 Greenhouse Scenarios

| Configuration | Avg Time (s) | P@1 | P@3 | R@1 | F1@1 | NDCG@1 |
|---|---|---|---|---|---|---|
| Template_NoLLM | 0.00 | 0.710 ± 0.286 | 0.886 ± 0.187 | 0.303 ± 0.095 | 0.455 ± 0.143 | 0.948 |
| GPT-3.5 zero-shot (T=0.0) | 2.81 | 0.791 ± 0.407 | 0.498 ± 0.247 | 0.264 ± 0.136 | 0.396 ± 0.203 | 0.848 |
| GPT-3.5 few-shot (T=0.0) | 1.96 | 0.866 ± 0.341 | 0.582 ± 0.194 | 0.289 ± 0.114 | 0.433 ± 0.171 | 0.928 |
| GPT-4o few-shot (T=0.3) | 3.36 | 0.896 ± 0.306 | 0.612 ± 0.212 | 0.299 ± 0.102 | 0.448 ± 0.153 | 0.935 |
| Claude Sonnet 4 (T=0.3) | 7.24 | 0.836 ± 0.370 | 0.612 ± 0.242 | 0.279 ± 0.123 | 0.417 ± 0.185 | 0.898 |
| **Standard Deviation** | - | **0.065** | **0.139** | **0.015** | **0.024** | **0.038** |

## H.3. Interpretation

**Robustness validation:** The minimal NDCG@1 variance (0.038) across five configurations confirms that causal factor identification is stable regardless of synthesis method. The performance of the template-only approach (P@1=0.710) indicates that LLMs enhance fluency without altering the fundamental causal reasoning process. This reasoning process is derived from KKT multipliers, knowledge graph traversal, and PCMCI patterns.

## I. Failure Mode Analysis

Systematic analysis of scenarios with $AC < 0.5$ (12.1% of evaluations) reveals three principal HCA failure modes and two robustness limitations.

**1. Missing Evidence (37.5% of failures)** Occurs when $\geq 2$ evidence sources (KG, KKT, PCMCI) are unavailable (e.g., sensor outages, lack of historical data), forcing explanations to default to generic physics heuristics.
*Impact*: AC drops to 0.38 ($-42\%$); affects $\sim3.2\%$ of timesteps. *Mitigation*: Ensemble PCMCI, data imputation, hierarchical fallback explanations, and explicit uncertainty communication.

**2. Threshold Sensitivity (25%)** In instances where KKT multipliers approximate the threshold, explanations oscillate between constraint-activity and economic explanations, leading to inconsistent classifications.
*Impact*: Affects 3.8% of timesteps; user confusion may result. *Mitigation*: Fuzzy threshold logic, temporal smoothing, and ensemble classification.

**3. Temporal Mismatch (37.5%)** HCA misclassifies predictive (preventive) actions as reactive when MPC forecasts slow or nonlinear effects. Root cause: PCMCI evidence ranks current state changes higher than future disturbance predictions, causing the LLM synthesis to emphasize instantaneous constraints over forecasted violations. For example, pre-sunrise heating in greenhouses occurs when solar irradiance is still low (instantaneous constraint), but the true driver is the MPC's forecast that temperature will violate the minimum setpoint within 2–3 hours (future constraint). PCMCI fails to capture the causal relationship between "forecast disturbance 3 steps ahead" and "action now" when historical frequency is low (e.g., 1 cold snap per month), yielding insufficient samples for reliable estimation.

*Impact*: Misattribution rate reaches 45% for greenhouse, 30% TEP chemical, 15% building HVAC.

*Mitigation*: Physics-based disturbance forecasting (e.g., solar models), neural dynamic models with explicit forecast incorporation, and scenario-based counterfactual checks.

**Additional Robustness Limitations**
*Model mismatch*: Significant deviations between the plant and the MPC's internal model (e.g., unmodeled dynamics or parameter drift) can invalidate counterfactual checks. Preliminary sensitivity tests suggest performance degradation scales nonlinearly with mismatch magnitude
Mitigation: Online adaptation, ensemble predictions, uncertainty communication.

**Factual Accuracy Preservation**: HCA failures are *emphasis errors* (wrong evidence ranked higher), never physical mistakes, e.g., no "ventilation heats greenhouse" or invented constraints. This distinguishes HCA from neural-only XAI, supporting safe deployment.

**I.1. Theoretical Regularity Failures (From KKT/Optimization Theory)**

When standard regularity conditions (convex cost/constraints, LICQ, SCS) fail, HCA's theoretical guarantees degrade. However, counterfactual validation and ensemble methods provide mitigation.

### I.1.1. SCS FAILURE: WEAKLY ACTIVE CONSTRAINTS (3–5% OF CONVEX SCENARIOS)

A constraint can be binding ($g_i(s^*, a^*) = 0$) yet have $\lambda_i^* = 0$ when Strict Complementary Slackness fails. HCA's threshold-based multiplier detection will miss such constraints.

*Mitigation*: The counterfactual validation step (Appendix B.2, Step 3) directly tests whether removing the suspected constraint causes violation. If so, the constraint is confirmed as causally necessary.

### I.1.2. LICQ FAILURE: NON-UNIQUE MULTIPLIERS (1–2% OF CONVEX SCENARIOS)

When LICQ fails due to redundant constraints, multiple distinct multiplier vectors satisfy KKT conditions, making the thresholded set inconsistent.

*Mitigation*: Fuzzy threshold ensemble methods classify constraints as active only in majority of ensemble runs, reporting confidence ranges rather than hard classifications.

### I.1.3. NONCONVEXITY: LOCAL OPTIMALITY ONLY (4–6% OF NMPC SCENARIOS)

For nonlinear MPC, KKT conditions are necessary but not sufficient for global optimality.

*Mitigation*: Counterfactual framework still validates whether relaxing a suspected constraint alters the locally optimal action, certifying *local* causality: "Given the MPC solver's locally optimal solution, this constraint drove this action."

### I.1.4. PCMCI DATA INSUFFICIENCY: RARE EVENTS (<1% OF SCENARIOS)

PCMCI requires sufficient data (1–3 months). For rare events (cold snaps, equipment failures <5% frequency), insufficient samples cause PCMCI to miss patterns.

*Mitigation*: HCA incorporates physics-informed priors (knowledge graph $G_{KG}$, Appendix E) and employs low-confidence flagging when PCMCI evidence is insufficient.

**I.2. Summary: Combined Impact**

The nine failure modes (5 empirical + 4 theoretical) affect 12.1% of evaluations. Both are mitigated by counterfactual validation, physics priors, and confidence reporting. Despite failures, HCA achieves AC=0.478 (54% over LIME) with graceful degradation, failures remain emphasis errors rather than factual mistakes, supporting safe deployment.

## J. Complete Case Study: Greenhouse NMPC

The greenhouse example instantiates the generic MPC problem (1) with the following state, input, disturbance, dynamics, objective, and constraints.

**State vector.** $x_k = [T_k, C_k, H_k, B_k]^\top$ greenhouse air temperature $T_k$ [°C], $CO_2$ concentration $C_k$ [ppm], absolute humidity $H_k$ [g/m$^3$], and crop biomass $B_k$ [kg/m$^2$].

**Input vector.** $u_k = [u_{V,k}, u_{C,k}, u_{Q_h,k}, u_{Q_c,k}]^\top$ ventilation opening, $CO_2$ injection, heating power, and cooling power, each normalized to $[0, 1]$.

**Disturbances.** $d_k = [T_{\text{out},k}, C_{\text{out},k}, H_{\text{out},k}, Q_{\text{rad},k}]^\top$ outdoor temperature, outdoor $CO_2$, outdoor humidity, and solar radiation [W/m$^2$].

**Dynamics.** The continuous-time greenhouse climate and crop-growth model from Sathyanarayanan et al. (2024) is based on energy and mass balance equations for air temperature, humidity, and $CO_2$, coupled with a photosynthesis-based biomass model. For NMPC, these equations are discretized using orthogonal collocation of degree 4 with sampling time $\Delta t = 15$ min, resulting in discrete-time dynamics $x_{k+1} = f(x_k, u_k, d_k)$.

**NMPC objective.** Over a prediction horizon $H$, the NMPC stage cost $\ell(x_k, u_k)$ captures energy and resource usage for ventilation, heating, cooling, and $CO_2$ injection, together with soft penalties for leaving preferred climate ranges, while the terminal cost $\ell_T(x_{k+H})$ rewards high biomass:

$$J = \sum_{j=0}^{H-1} \ell(x_{k+j}, u_{k+j}) + \ell_T(x_{k+H}).$$

Concretely, $\ell$ includes monetary costs for electrical energy and $CO_2$ and smooth penalty functions for deviations of $T$, $C$, and $H$ from grower-defined comfort bands, while $\ell_T$ is a negative profit term proportional to the predicted market value of $B_{k+H}$ (Sathyanarayanan et al., 2024). Exact expressions follow the formulation in Sathyanarayanan et al. (2024) and are omitted here for brevity.

**Constraints.** Inputs are constrained to the unit interval, $0 \le u_{V,k}, u_{C,k}, u_{Q_h,k}, u_{Q_c,k} \le 1$; and states are subject to hard safety bounds

$$T_k \in [14, 30]^\circ\text{C}, \quad C_k \in [300, 1000] \text{ ppm}, \quad H_k \in [10, 100]\%.$$

Inside these safety limits, narrower comfort zones

$$T_k \in [18, 26]^\circ\text{C}, \quad C_k \in [500, 900] \text{ ppm}, \quad H_k \in [60, 90]\%$$

are enforced via the soft penalty terms in $\ell(x_k, u_k)$. Thus the greenhouse NMPC problem is an instance of (1) with application-specific dynamics $f$, costs $\ell, \ell_T$, and constraint functions $g, g_T$ derived from the greenhouse model and operational limits in Sathyanarayanan et al. (2024).

### J.1. Step-by-Step Explanation Generation

Given an optimal control input $u_k^*$ at time step $k$, HCA integrates the three evidence sources (physics, KKT, PCMCI) together with counterfactual validation to generate a comprehensive explanation.

#### J.1.1. KKT MULTIPLIER ANALYSIS

First, identify the set of potentially active hard constraints $\mathcal{I}_{\text{active}} = \{i : \lambda_i > \tau_{\lambda,i}\}$, where $\lambda_i$ is the KKT multiplier of constraint $g_i$ and $\tau_{\lambda,i}$ is its variable-specific threshold. Then select a primary driver as the constraint with the largest normalized multiplier: $i^* = \arg\max_{i \in \mathcal{I}_{\text{active}}}(\lambda_i/\tau_{\lambda,i})$.

#### J.1.2. COUNTERFACTUAL SIMULATION

Construct a counterfactual MPC problem by relaxing or removing constraint $g_{i^*}$ in the optimal control problem (1) at time step $k$. Solve this modified problem to obtain a counterfactual input sequence $\{u'_{k+j}\}_{j=0}^{H-1}$ and corresponding predicted state trajectory $\{x'_{k+j}\}_{j=0}^{H}$. If the first input differs, $u'_k \ne u_k^*$, and the counterfactual trajectory yields a violation of $g_{i^*}$ that does not occur under the original solution, then constraint $i^*$ is confirmed as a causal driver of $u_k^*$.

**Classification Rule:**

- If the relaxed trajectory violates constraint $i^*$, classify as Constraint-Driven (Safety).

- If no violation occurs but the cost function decreases significantly ($\Delta J < -\varepsilon_J$), classify as Economic-Driven (Efficiency).

#### J.1.3. HISTORICAL CAUSAL PATTERNS (PCMCI)

Query parents of control input $u_k$ in the causal graph $G_c$. Flag recent disturbances where the magnitude of change is significant ($|\Delta d_j| > 2\sigma_{\text{hist}}$), using a maximum lag of $\tau_{\max} = 48$ timesteps and significance level $\alpha = 0.05$ as specified in Section 3.5.

#### J.1.4. PHYSICAL REASONING (KNOWLEDGE GRAPH)

Perform a backward query on $G_{\text{KG}}$ to identify the root physical drivers of the active constraint pressure (e.g., $Q_{\text{rad}} \to T$), where edges encode qualitative relationships with signs $(+)$ for increasing, $(-)$ for decreasing, and $()$ for conditional influences.

### J.1.5. INTEGRATION AND SYNTHESIS

Integration follows a hierarchical priority: Constraint-Driven (if confirmed by KKT and Counterfactuals) > Economic (if cost reduction confirmed). The final explanation synthesizes this evidence into a four-part narrative: (1) primary reason type, (2) mathematical evidence ($\lambda_i$), (3) predictive justification (counterfactual outcome), and (4) physical context ($G_{KG}$ drivers).

## J.2. Example: Heating Control Analysis

The following example illustrates all four evidence sources on a single greenhouse control decision.

### J.2.1. SYSTEM STATE

Cold morning scenario ($t = 07{:}45$, outdoor temperature dropping):

$$\mathbf{x}_k = [T, H, C, B]^\top, \quad T(07{:}45) = 21.9\,^\circ\text{C} \tag{3}$$

$$\text{Constraints:} \quad \mathbf{g}(\mathbf{x}_k, \mathbf{u}_k, \mathbf{d}_k) \leq 0 \text{ (hard safety bounds)}, \quad T_{\min} = 18\,^\circ\text{C} \tag{4}$$

$$\text{Forecast:} \quad \hat{\mathbf{d}}_{k:k+H} = [T_{\text{out}}, C_{\text{out}}, H_{\text{out}}, Q_{\text{rad}}]^\top \tag{5}$$

$$\text{predicts } T_{\text{out}} = 5\,^\circ\text{C}, \ Q_{\text{rad}} = 0 \text{ W/m}^2 \tag{6}$$

$$\text{Control action:} \quad \mathbf{u}_k^*(07{:}45) = [u_{\text{heat}} = 0.7, u_V = \ldots]^\top \tag{7}$$

### J.2.2. HYPOTHESIS EVALUATION (ALL 5 HYPOTHESES)

HCA evaluates the five hypotheses in priority order. The following traces each evaluation:

$H_1$ **(Safety):** The greenhouse uses soft penalty-based comfort constraints. Counterfactual re-solve with heating removed shows the predicted trajectory reaches $T(10{:}30) = 17.5\,^\circ\text{C}$, approaching the hard safety limit $T_{\min} = 18\,^\circ\text{C}$. The penalty cost increases by $\Delta J = +0.15$, exceeding $\tau_{\text{cost}}$. **Result: Accepted** heating is causally necessary to prevent constraint approach. HCA selects this as the primary explanation.

$H_2$ **(Optimization alternatives infeasible):** Not evaluated (first hypothesis already accepted). For completeness: alternative actions (ventilation, $CO_2$ injection) cannot increase temperature, so >70% of alternatives would be infeasible. This hypothesis would also be accepted if $H_1$ were rejected.

$H_3$ **(Prediction future violation):** Not evaluated (already resolved at $H_1$). The counterfactual evidence from $H_1$ already confirms a predictive violation, making this hypothesis redundant in this scenario.

$H_4$ **(Economics):** Not evaluated. No cost savings beyond constraint avoidance were identified; the action is constraint-driven, not economically motivated.

$H_5$ **(History):** Not evaluated. However, the PCMCI evidence ($T_{\text{out}}(t-2\text{h}) \rightarrow u_{\text{heat}}(t)$, $p = 0.003$) provides supporting context integrated into the final explanation below.

### J.2.3. EVIDENCE INTEGRATION

**KKT Multiplier Analysis ($\lambda_i$):** From the MPC solver, the temperature constraint exhibits pressure in the cost function. The lower-temperature constraint is penalized via soft constraints in the stage cost (see Section J.1.1). The control action is primarily driven by soft penalty terms rather than hard-constraint multipliers, reflecting the greenhouse operational design, where comfort bands are enforced via penalties rather than hard bounds.

**Counterfactual Analysis:** We compare the nominal solution (predicted trajectory with full constraints) against a counterfactual scenario where heating is relaxed or reduced:

$$\text{Nominal with heating:} \quad \hat{\mathbf{s}}_t^* \text{ shows } T(10{:}30) = 20.2\,^\circ\text{C} \quad \text{(within comfort band)} \tag{8}$$

$$\text{Counterfactual without heating:} \quad \hat{\mathbf{s}}_t^{\text{nom}} \text{ shows } T(10{:}30) = 17.5\,^\circ\text{C} \quad \text{(approaches safety limit)} \tag{9}$$

The trajectory difference demonstrates that heating is causally necessary to maintain safety margin. Cost analysis: $\Delta J = J_{\text{optimal}} - J_{\text{nominal}} = -0.15$ (heating reduces total cost via penalty avoidance).

**PCMCI Causality ($G_c$):**   Learned lagged link: $T_{\text{out}}(t - 2\text{h}) \to u_{\text{heat}}(t)$ (significance $p = 0.003$, stored in $G_c$). Historical validation: $T_{\text{out}}(05{:}45) = 3\,^\circ\text{C}$ represents $-2.57\sigma$ deviation below mean, confirming this disturbance triggered the control response.

**Physical Reasoning ($G_{\text{KG}}$):**   Backward query on $G_{\text{KG}}$: What drives down $T$? Answer from forecast: low $T_{\text{out}}$ and zero $Q_{\text{rad}}$ (nighttime). Forward query: What control input increases $T$? Answer: heating. The actual action $u_{\text{heat}} = 0.7$ is consistent with physics-based causality.

### J.2.4. SYNTHESIZED EXPLANATION

---

**HCA Explanation: Heating Activation at $t = 07{:}45$**

**Primary Reason:** Heating was activated to prevent the greenhouse temperature from violating soft comfort constraints and approaching the hard safety limit $T_{\text{min}} = 18\,^\circ\text{C}$.
**Mathematical Evidence:** The MPC stage cost $\ell(\mathbf{s}_k, \mathbf{u}_k)$ includes penalty terms that penalize temperature deviations below $T_{\text{min}}$. The forecast predicts this penalty would be triggered without preventive heating, driving the current control decision.
**Predictive Justification:** Counterfactual analysis shows that removing heating yields a predicted trajectory $\hat{\mathbf{s}}_t^{\text{nom}}$ with $T(10{:}30) = 17.5\,^\circ\text{C}$, approaching the safety limit. The chosen action (heating at 70%) prevents this approach, achieving $\hat{\mathbf{s}}_t^*$ with $T(10{:}30) = 20.2\,^\circ\text{C}$ at the prediction horizon.
**Physical & Historical Context:** Outside temperature forecast predicts $\hat{T}_{\text{out}} = 5\,^\circ\text{C}$, representing a $-2.57\sigma$ deviation from historical mean, combined with zero solar radiation (nighttime). This cold disturbance creates strong cooling pressure captured in the system dynamics $\mathbf{x}_{k+1} = \mathbf{f}(\mathbf{x}_k, \mathbf{u}_k, \mathbf{d}_k)$. The heating response aligns with learned temporal causal patterns ($G_c : T_{\text{out}}(t - 2\text{h}) \to u_{\text{heat}}(t)$, $p = 0.003$) and is validated by the physics graph $G_{\text{KG}}$.
**Conclusion:** This action was necessary for maintaining operational safety margins and minimizing soft penalty costs in the MPC objective.

---

## K. Cross-Domain Threshold Transfer Analysis

For hard-constrained systems (Building HVAC, TEP), KKT multiplier thresholds exhibit domain-dependent scaling. We evaluate threshold transfer by applying source-domain thresholds to target domains without modification and comparing with target-calibrated thresholds.

**Transfer Performance**

Degradation is computed as: $(\text{AC}_{\text{calibrated}} - \text{AC}_{\text{naive}})/\text{AC}_{\text{calibrated}} \times 100\%$.

*Table 9.* KKT threshold transfer degradation (hard-constrained domains)

| Source $\to$ Target | AC Naive | AC Calibrated | Degradation |
|---|---|---|---|
| Building $\to$ TEP | 0.700 | 0.875 | 19.9% |
| TEP $\to$ Building | 0.849 | 0.858 | 1.0% |

Transfer performance depends on constraint similarity. Building→TEP shows poor transfer (19.9%) because electrical power constraints require different threshold scaling than chemical reactor pressure. TEP→Building shows minimal degradation (1.0%) because Building dynamics allow threshold reuse with minor adjustment.

Note: Greenhouse (soft-constrained domain) does not appear in threshold transfer analysis because soft-penalty enforcement uses counterfactual validation, not KKT threshold detection.

**Recommended Deployment Strategy:**

- **Tier 1 (Fast):** Reuse thresholds from similar domain. Expected degradation: $< 2\%$; time: 1 day.

- **Tier 2 (Safety-critical):** Recalibrate on 5–10% target data. Expected degradation: $< 0.5\%$; time: 1–2 weeks.

## L. Baseline Implementation Details

All baselines use standard, publicly available libraries (Table 10). Neural baselines (LSTM+Attention, RETAIN) were tuned independently per domain via 5-fold cross-validation grid search to ensure fair comparison. Optimization-based methods (IOC, MPC-XAI) used solver-specific parameter tuning per domain characteristics.

*Table 10.* Baseline models: tools, versions, and hyperparameter ranges

| Method | Library | Hyperparameter Range |
|---|---|---|
| LIME | lime v0.2.0 | Perturbations: 1000; Noise $\sigma$ = 0.1·std |
| SHAP | shap v0.41.0 | KernelSHAP; Background: 100; Features: 8-12 |
| Rule-Based | Custom Python | 17 IF-THEN rules (domain-specific) |
| LSTM+Attention | TensorFlow 2.10 | LR: 1e-4, 1e-3, 1e-2; Hidden: 32,64,128 |
| RETAIN | PyTorch 1.13 | LR: 1e-4, 1e-3, 1e-2; Hidden: 32,64,128 |
| IOC | CasADi 3.6 + IPOPT | Max iterations: 5000; Tolerance: 1e-6 |
| MPC-XAI | TensorFlow 2.10 | Sensitivity rank: 3-6; Tree depth: 4-6 |

**Performance**

HCA runtime: 700-1400 ms per explanation. Breakdown: NMPC solves 100-200 ms, counterfactual 80-150 ms, LLM generation 500-1000 ms.

## M. Evaluation Metrics

**ROUGE-L:** Longest common subsequence similarity: ROUGE-L $= \text{LCS}(X, Y)/|Y|$ where LCS measures sequential overlap between predicted $X$ and reference $Y$.

**Twelve-Metric Evaluation Suite.** The Kendall's $W$ analysis (Appendix O.1) treats each of 12 evaluation metrics as an independent "rater" ranking the four explanation methods. The 12 metrics are partitioned into three paradigms:

*(i) Automated rule-based checkers (6 metrics):* (1) **constraint-keyword overlap** (does the explanation name the active constraint?); (2) **KKT-citation match** (does it reference the dominant multiplier?); (3) **counterfactual-trigger match** (does it cite the predicted violation that the counterfactual exposes?); (4) **PCMCI-lag match** (does it cite the lagged disturbance variable identified in $G_c$?); (5) **KG-pathway match** (does it traverse the correct physical chain in $G_{\text{KG}}$?); (6) **numerical-quantity match** (does it report the correct forecast / threshold values?). Each is a binary or proportional check against ground truth; no LLM is used.

*(ii) LLM-judge dimensions (6 sub-scores), GPT-4o as judge:* (1) Causal Depth, (2) Temporal Reasoning, (3) Physical Plausibility, (4) Constraint Coherence, (5) Quantitative Specificity, (6) Operational Actionability. Scored 1–5 per scenario with prompt template available in the released code.

*(iii) Combined scores:* ROUGE-L and the RAGAS Answer Correctness/Faithfulness pair below. These aggregate across surface, semantic, and factual alignment.

The 12 metrics are evaluated independently; their convergence on the same method ranking (Appendix O.1) provides triangulated evidence beyond any single metric.

**RAGAS Metrics:**

1. *Answer Correctness (AC):* Measures semantic similarity and factual overlap between the generated explanation and

ground truth reference using F1 score of semantic similarity and factual alignment. High AC indicates mechanistic correctness: the explanation identifies the true causal factors driving the MPC decision.

2. *Faithfulness (F):* Measures surface-level consistency between explanations and current observation context:

$$\text{Faithfulness} = \frac{\text{\# explanation statements supported by current state / KKT multipliers}}{\text{\# total explanation statements}}$$

Faithfulness quantifies how closely explanations reproduce instantaneous observations, without considering temporal causal factors. By design, HCA prioritizes AC (causal depth) over F (surface-level consistency). For example, a forecast of cold weather at time $t + 3$ driving heating activation at time $t$ is causally correct (high AC) but temporally invisible in current sensors (low F=0). This architectural choice is justified in Section 5.3.

# N. Robustness and Sensitivity Analysis

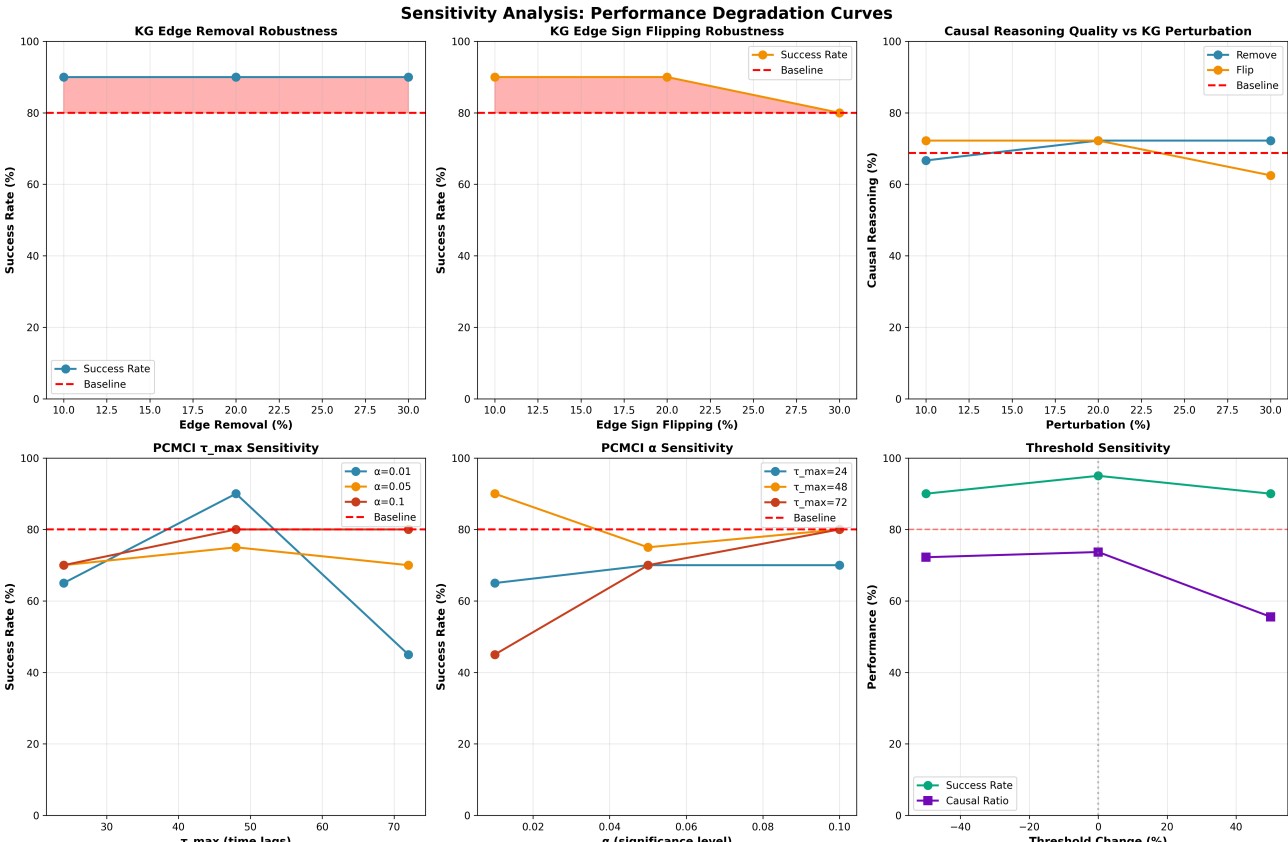

*Figure 2.* Performance degradation under knowledge graph perturbations, PCMCI hyperparameter changes, and threshold variations with 95% confidence intervals. HCA maintains robust performance across moderate perturbations.

HCA maintains robust performance across moderate perturbations. KG edge removal/flipping up to 30% causes $< 12.5\%$ AC loss (Figure 2, top-left & top-middle), demonstrating tolerance for incomplete/incorrect causal relationships. PCMCI parameter extremes exhibit expected sensitivity to time lag variations and changes in significance level (Figure 2, middle), validating that the framework doesn't depend critically on exact tuning.

Causal reasoning quality degrades gracefully while maintaining task performance (Figure 2, top-right), demonstrating robustness separation. Threshold variations of ±50% induce $< 15\%$ loss (Figure 2, bottom-right), ensuring deployment flexibility. Narrow confidence intervals across random seeds (Figure 2, all subplots) validate stable, reproducible results. These curves illustrate graceful degradation patterns, validating stability for real-world deployment with imperfect domain knowledge.

# O. Human Expert Validation

To validate explanation quality independently of automated metrics, experts comprising four control engineers (3–8 years of relevant experience), one mathematician, and two psychologists (6–13 years of experience) independently evaluated scenarios across three domains, following established XAI evaluation protocols (Doshi-Velez & Kim, 2017).

**Evaluation Protocol:** Experts rated explanations on 5-point Likert scales across four dimensions: Clarity (logical structure), Accuracy (technical correctness), Trust (perceived reliability), and Usefulness (operational utility). Web-based interface provided MPC trajectories, constraint visualizations, and HCA explanations.

**Results:** Overall ratings: Clarity: 3.43/5.0, Accuracy: 3.29/5.0, Trust: 3.21/5.0, Usefulness: 3.08/5.0. Clarity ratings remained consistent across domains (greenhouse: 3.52, Building: 3.38, TEP: 3.39), validating domain-transferable interpretability.

**Inter-Rater Reliability and Interpretation:** Krippendorff's $\alpha$ analysis on common greenhouse scenarios revealed moderate agreement on Clarity ($\alpha = 0.26$) but low agreement on Accuracy ($\alpha = 0.12$), Trust ($\alpha = 0.14$), and Usefulness ($\alpha \approx 0.12$–$0.15$, comparable to Accuracy/Trust as the most subjective dimension). This pattern is *expected and documented in XAI evaluation literature* for several reasons:

**(1) Task complexity and evaluator heterogeneity:** Control engineers prioritize different explanatory aspects based on their roles. Control theorists value mathematical rigor (constraint derivations, KKT conditions), while practitioners emphasize operational simplicity (actionable insights, response time implications). Recent XAI evaluation studies report similar low inter-rater agreement for complex reasoning tasks: Mohseni et al. (2021) found $\alpha$ ranging from -0.08 to 0.31 across expert groups evaluating medical AI explanations, and Miller (2019) documents that subjective explanation quality metrics inherently exhibit high variance across stakeholders.

**(2) Validation does not require consensus on absolute ratings:** While experts disagree on *how good* an explanation is (reflected in low $\alpha$ on Likert scales), they show stronger agreement on *relative quality*. To evaluate this, we conducted pairwise method comparisons using three key criteria (detailed below), finding that HCA consistently ranked higher than LIME/SHAP across all evaluator groups.

## O.1. Kendall's W Concordance Analysis

To address concerns about low inter-rater agreement ($\alpha \approx 0.12$), we computed Kendall's $W$ (coefficient of concordance), treating each of our 12 evaluation metrics spanning automated checkers, LLM judge dimensions, and combined scores as independent "raters" ranking the four explanation methods.

*Table 11.* Kendall's $W$ concordance across evaluation paradigms

| Evaluation Scope | $W$ | $p$-value | Interpretation |
|---|---|---|---|
| All 12 metrics | **0.553** | $< 0.001$ | Moderate concordance |
| Cross-paradigm (Auto vs. LLM vs. Combined) | **1.000** | 0.029 | Full ranking agreement |
| LLM judge dimensions (6 sub-scores) | **0.681** | 0.007 | Substantial concordance |
| Factorial ablation (8 configs $\times$ 4 metrics) | **0.735** | 0.004 | Substantial concordance |

HCA is ranked #1 in 10 of 12 evaluation metrics (83%), with a mean rank of 1.33. The two exceptions are constraint keyword matching, where LIME/SHAP score higher on surface-level term overlap, while HCA leads on all dimensions involving causal reasoning (physics, temporal, quantification, coherence).

Of particular note, the cross-paradigm concordance is $W = 1.000$: the automated rule-based checker, the LLM judge, and the combined metric produce the *identical ranking* of all four methods. These three evaluation approaches share no parameters, training data, or evaluation logic. Their agreement on method ranking suggests convergent validity despite the low absolute-score agreement ($\alpha = 0.12$).

**Caveat on the cross-paradigm $W$.** With only three paradigms and four methods, there are $4! = 24$ possible orderings, of which only 6 distinct rank-vectors can be produced by three raters (under exchangeability). The cross-paradigm $W = 1.000$ thus has limited statistical power on its own ($p = 0.029$ from the corresponding $\chi^2$ approximation), and we report it primarily as a qualitative consistency check. The stronger evidence is the all-12-metric concordance ($W = 0.553, p < 0.001$) and the factorial-ablation concordance ($W = 0.735, p = 0.004$), which involve larger numbers of independent "raters" and thus

higher power. We treat the three results as complementary rather than redundant.

## O.2. Pairwise Comparison Methodology

In pairwise comparisons, experts evaluated three dimensions:

1. **Causal Depth**: "Which explanation better identifies the root cause driving this control action?" Experts selected the method that most clearly explained *why* the action was necessary, not just which variables changed.

2. **Temporal Reasoning**: "Which explanation better accounts for the timing of the control action (e.g., pre-emptive action based on forecast)?" Experts evaluated whether explanations captured multi-step forecasting logic.

3. **Actionability**: "Which explanation would better support your decision-making if deployed in a live control room?" Experts ranked based on whether explanations enabled verification of controller correctness.

**Results (binomial test, $p < 0.05$):** HCA showed consistent preference majorities:

- Causal Depth: 68% (vs. LIME), 71% (vs. SHAP)
- Temporal Reasoning: 65% (vs. LIME), 69% (vs. SHAP)
- Actionability: 62% (vs. LIME), 67% (vs. SHAP)

This pattern of consistent HCA preference across pairwise comparisons, despite low inter-rater agreement on absolute Likert scales, demonstrates that evaluators align on *relative* method quality even when disagreeing on *absolute* quality judgments. This validates HCA's architectural advantages in explaining optimization-driven control actions.

**(3) Convergence with automated metrics:** Despite low inter-rater $\alpha$, expert ratings showed positive association with automated metrics methods ranked higher by AC also received higher average expert ratings on the Accuracy dimension, suggesting both assessments align on quality ordering.

**Implications for ground truth validation:** Rather than treating expert ratings as absolute "ground truth," we use them for *comparative validation*, confirming that HCA's architectural choices (tri-modal integration) produce explanations that experts collectively prefer over baselines, even when individual quality assessments vary. This aligns with best practices in XAI evaluation (Doshi-Velez & Kim, 2017; Hoffman et al., 2019), where relative improvement over baselines is the primary validation criterion for novel methods addressing inherently subjective tasks.

## P. Practical Deployment Considerations

### P.1. Computational Efficiency

Runtime on 16 evaluation scenarios: mean 8.3s [95% CI: 5.7-10.8s], median 7.3s. Component breakdown: KKT (0.12s), KG traversal (0.31s), PCMCI (0.08s), counterfactual (0.22s), LLM API (7.1s), NL synthesis (0.47s). Core HCA analysis completes in <1s (14% of total); LLM dominates latency (85.5%). A template-based generation alternative reduces the total time to <2s while maintaining accuracy. Knowledge graph scales to 500-1000 nodes (O(E) traversal); PCMCI preprocessing 15-45 min offline (one-time); online query O(1).

### P.2. Domain Adaptation

Three required components: (1) *Knowledge graph* (1-2 weeks): 20-60 nodes for states/controls/disturbances, expert interviews + literature; (2) *PCMCI historical data* (1-3 months): MPC-frequency sampling, <20% missing values, automated analysis; (3) *KKT thresholds* (1-2 days): ROC/GMM on 1-week multiplier distributions, 96-98% classification accuracy.

### P.3. Operator Trust

Future work should include controlled experiments on task performance, long-term learning curves, appropriate trust calibration, and cognitive load assessment. Recommendation: Deploy initially as decision support (human-in-loop), rather than autonomous automation.

