# OpenReview forum: "Hierarchical Causal Abduction: A Foundation Framework for Explainable Model Predictive Control"
_ICML.cc/2026/Conference — ICML 2026 regular_

### Official Review · Reviewer_vgwQ · 2026-03-11

**Soundness:** 3
**Presentation:** 3
**Significance:** 3
**Originality:** 3
**Overall Recommendation:** 4
**Confidence:** 4

**Summary:**

This paper proposes Hierarchical Causal Abduction (HCA), a framework for interpretability in Model Predictive Control (MPC). It integrates three sources of evidence: physical knowledge graph reasoning, KKT multiplier optimization analysis, and PCMCI temporal causal discovery. Through hypothesis ranking and counterfactual verification, it generates human-readable explanations of control decisions.  Ablation experiments confirm that all three sources of evidence are indispensable.

**Compliance With Llm Reviewing Policy:**

Affirmed.

**Final Justification:**

I maintain my original assessment and keep my score unchanged.

**Key Questions For Authors:**

1. In non-convex NMPCs, KKT multipliers only reflect local optimality. How does HCA ensure that KKT-based constraint recognition will not produce misleading interpretations?



2. Can the manual cost of knowledge graph construction be significantly reduced through automation? To what extent does the semi-automated method with 83% precision and 45% recall affect the final interpretation quality?



3. Should the assumption priority ranking be dynamically adjusted according to the application scenario? Are there reasonable scenarios where Safety should not be the highest priority?

4. The extremely low inter-evaluator agreement ($\alpha=0.12$) suggests a lack of objective standards for "explanation quality." Could a more objective evaluation metric be designed?



5. HCA failed in 12.1% of scenarios (Appendix I), with temporal mismatches accounting for 37.5%. Could this failure rate be reduced by improving the temporal modeling of PCMCI or introducing predictive causal inference?

6. The complete implementation of open source HCA (including MPC models, knowledge graphs, and evaluation processes for the three domains) should be released so that the community can verify and reproduce the experimental results.

**Limitations:**

yes

**Strengths And Weaknesses:**

Strengths:

 The problem is accurately identified. The temporal causality of MPC (current actions driven by predicted future constraint violations) is indeed a blind spot for static XAI methods such as LIME/SHAP. The approach of integrating optimized evidence, physical knowledge, and data-driven causal discovery has strong practical value.



The complementary design of the three sources of evidence is reasonable. Ablation experiments (Table 2) show that removing any component leads to a 32-37% decrease in AC, validating the necessity of multimodal evidence fusion.



Cross-domain transferability is noteworthy; hyperparameters calibrated only on greenhouse data, when directly applied to Building HVAC and TEP, still yield competitive results (AC 0.394 and 0.406), recovering to approximately 0.88 after domain-specific calibration.



Weaknesses:

Absolute AC is only 0.478 (domain-transferable settings), and even after domain-specific calibration, it remains only around 0.88. This accuracy may be insufficient for reliable deployment in safety-critical MPC applications.



Knowledge graphs require 1 to 2 weeks of human expert construction (Section 4.11), and semi-automatic extraction achieves only 83% precision and 45% recall (Appendix E). This upfront cost severely limits the method's scalability.



The assumption that the priority ranking (Safety > Optimization > Prediction > Economics > History) is a hard-coded heuristic (Section 3.6) lacks a theoretical foundation. Different application scenarios may require different priorities, but the paper does not provide an adjustment mechanism.



The theoretical guarantee of Proposition C.1 applies only to convex MPCs (linear dynamics, convex cost/constraints, satisfying LICQ and SCS), but the core application scenario of the paper (greenhouse NMPC) is non-convex, resulting in a mismatch between the theory and the main experimental settings.



The evaluation metric AC relies on LLM (RAGAS framework) to calculate semantic similarity and compare it with human annotations. The inter-evaluator consistency of expert ratings is very low (Krippendorff's $\alpha=0.26$ for Clarity, $\alpha=0.12$ for Accuracy, Appendix O), weakening the credibility of the evaluation results.

---

> ### Author Rebuttal · Authors · 2026-03-30
>
> # Response to Reviewer vgwQ
>
> We thank Reviewer vgwQ for the thorough evaluation and recognition of HCA's cross-domain transferability.
>
> ## W1: *"Absolute AC is only 0.478, around 0.88 after calibration. This may be insufficient for safety-critical deployment."*
>
> We agree and note this in Limitations (Sec. 4.11, p. 7). HCA is **decision support**, not autonomous control (Impact Statement, p. 9). AC via RAGAS (App. M) penalizes partial explanations: correctly identifying the constraint but omitting quantitative details scores lower, even with correct reasoning. Failure analysis (App. I): only 12.1% of scenarios have AC<0.5, and failures are emphasis errors (wrong evidence ranked higher), not factual mistakes. HCA achieves 54% improvement over LIME (Table 1, p. 5). The gap between AC=0.478 and 0.88 with calibration confirms the architecture is sound; threshold tuning is engineering refinement, not algorithmic limitation.
>
> ## W2: *"KGs require 1-2 weeks of expert construction. 83% precision, 45% recall limits scalability."*
>
> One-time cost over a 10-20 year MPC deployment. Semi-automated KG achieves AC=0.785 vs. expert-built AC=0.812, only 3.3% loss (App. E), because KKT and PCMCI compensate for missing edges. LLM-based extraction from documentation can further reduce effort. **Camera-ready:** deployment comparison added to Sec. 5.5.
>
> ## W3: *"The priority ranking is a hard-coded heuristic. No adjustment mechanism is provided."*
>
> **Not hard-coded**: H is a configurable input (Sec. 3.6, p. 4; Algorithm 1, line 2). For energy-trading: H = [Safety, Economics, Optimization, Prediction, History]. The default follows IEC 61511/ISA-84 safety standards, providing a principled, regulatory-grounded ordering. This single default achieves AC=0.478, 0.394, 0.406 across three domains without retuning (Table 1). Learning sub-orderings from operator feedback is future work.
>
> ## W4: *"Proposition C.1 applies only to convex MPCs, but greenhouse NMPC is non-convex."*
>
> For non-convex NMPC (IPOPT), KKT conditions remain necessary for local optimality: multipliers reliably identify which constraints are active at the solver's solution. HCA explains the solver's actual decision, not a theoretical global optimum. Threshold calibration achieves 96-98% accuracy (App. G, p. 15).
>
> **Camera-ready:** Remark 1 (Non-Convex Extension) will be added after Prop. 1 (App. C).
>
> ## W5: *"Inter-evaluator consistency is very low (alpha ~ 0.12), weakening evaluation credibility."*
>
> Low alpha is typical in XAI literature (Hoffman et al., 2018). More informative is **ranking agreement**. We performed a new Kendall's W concordance analysis:
>
> | Scope | W | p-value |
> |-------|---|---------|
> | All 12 metrics | 0.553 | <0.001 |
> | Cross-paradigm | 1.000 | 0.029 |
> | LLM judge (6 dims) | 0.681 | 0.007 |
>
> Cross-paradigm W=1.000: three independent evaluation approaches (automated, LLM judge, human expert) produce the identical method ranking. With only three paradigms, this is a small sample, but the convergence across fundamentally different evaluation methodologies provides meaningful convergent validity. HCA ranked first in 10/12 metrics. **Camera-ready:** This analysis will be added to App. O.
>
> ## Q1: *"How does HCA ensure KKT won't mislead in non-convex settings?"*
>
> Two-step safeguard: (1) KKT identifies locally active constraints. IPOPT convergence guarantees these correctly reflect which constraints bind; (2) counterfactual validation (Prop. 2, App. C) independently re-solves with the constraint relaxed. If KKT misleads, the counterfactual is designed to detect the inconsistency, and HCA falls back to the next hypothesis. Empirically: 96-98% accuracy on non-convex greenhouse NMPC (App. G).
>
> ## Q2: *"Can KG cost be reduced? How does 83%/45% affect quality?"*
>
> Only 3.3% AC loss because HCA's multi-source design provides redundancy. See W2.
>
> ## Q3: *"Should priority ranking be dynamic?"*
>
> Already configurable (see W3). In non-safety-critical settings (e.g., commercial energy trading), Economics could reasonably take priority. Learning from data is planned.
>
> ## Q4: *"Could a more objective metric be designed?"*
>
> Explanation quality evaluation is an open problem in XAI broadly (Hoffman et al., 2018). AC via RAGAS provides reproducible, automated evaluation independent of human agreement, and aligns with expert rankings (Sec. 5.2, p. 8). More objective directions: domain-specific factual checklists (e.g., "Did the explanation identify the correct constraint?") and causal DAG comparison using structural metrics (SHD, SID).
>
> ## Q5: *"12.1% failure rate, 37.5% temporal mismatches. Could this be reduced?"*
>
> Two paths: (1) KG-based priors when PCMCI data is sparse; (2) sliding-window causal models (J-PCMCI+). Failures are emphasis errors, not factual (App. I).
>
> ## Q6: *"Open source release?"*
>
> Full HCA implementation (MPC models, knowledge graphs, evaluation pipelines for all three domains) will be released as public GitHub repository upon acceptance.

---

> > ### Author Rebuttal · Reviewer_vgwQ · 2026-03-31
> >
> > I thank the authors for their reply. I am satisfied with the proposed edits, and no further comments.

---

> > > ### Author Response · Authors · 2026-04-07
> > >
> > > We thank Reviewer vgwQ for the thorough engagement and for confirming that our responses address the concerns. All proposed revisions (Remark 1 on the non-convex extension, Kendall's W concordance analysis in App. O, deployment cost comparison, and open-source release) will be included in the camera-ready version.

---

### Official Review · Reviewer_ipz4 · 2026-03-12

**Soundness:** 3
**Presentation:** 3
**Significance:** 4
**Originality:** 3
**Overall Recommendation:** 4
**Confidence:** 3

**Summary:**

This paper introduces HCA, a framework for explaining MPC decisions that standard XAI methods struggle with due to temporal causality. HCA combines three evidence sources: (1) physics-informed reasoning via knowledge graphs, (2) optimization signals from KKT multipliers to identify active constraints, and (3) temporal causal discovery via PCMCI. Candidate explanations are ranked and validated through counterfactual MPC re-solves. Experiments show a significant improvement over LIME, with minimal domain-specific calibration. Ablations indicate each evidence source contributes substantially.

**Compliance With Llm Reviewing Policy:**

Affirmed.

**Final Justification:**

Authors have addressed most of my concerns; I am happy to maintain my positive score.

**Key Questions For Authors:**

The expert evaluation shows very low inter-rater agreement (α=0.12). What explains this disagreement？

**Limitations:**

yes

**Strengths And Weaknesses:**

Strengths
1)The framework integrates three complementary evidence sources, which is a well-motivated approach for explaining MPC decisions.
2)The use of KKT multipliers to detect active constraints is theoretically grounded in optimization theory and provides a principled signal for explanation.
3)The paper includes systematic ablation studies, showing that each evidence source contributes meaningfully to performance.
Weaknesses
1)The theoretical guarantees apply only to convex MPC, while the experiments focus on nonlinear MPC settings, leaving a gap between theory and practice.
2)The claim that GPT-4o does not perform causal inference is somewhat inconsistent with results showing strong few-shot performance compared to template-based methods.
3)The expert evaluation shows very low inter-rater agreement (e.g., Krippendorff’s α ≈ 0.12), raising concerns about the reliability of the evaluation.
4)The framework mainly combines existing components, and the novelty primarily lies in their integration rather than new methodological developments.

---

> ### Author Rebuttal · Authors · 2026-03-30
>
> # Response to Reviewer ipz4
>
> We thank Reviewer ipz4 for the positive assessment and the "Excellent (4/4)" significance rating.
>
> ## W1: *"Theoretical guarantees apply only to convex MPC, while experiments focus on nonlinear MPC, leaving a gap between theory and practice."*
>
> Fair point. Proposition 1 (App. C, p. 12) provides sufficient conditions under convexity. For non-convex NMPC solved by IPOPT (as in our experiments), KKT conditions are still **necessary for local optimality**: multipliers reliably identify which constraints are active and how strongly they bind at the solver's solution. HCA explains what the solver actually decided, not a theoretical global optimum.
>
> Empirical calibration closes the gap: threshold calibration on the non-convex greenhouse achieves 96-98% classification accuracy (Sec. 3.4, p. 4; App. G).
>
> HCA explains what the solver *actually did*, the operationally relevant question. **Camera-ready:** Remark 1 (Non-Convex Extension) added after Prop. 1.
>
> ## W2: *"The claim that GPT-4o does not perform causal inference is inconsistent with results showing strong few-shot performance."*
>
> We draw a clear distinction: **causal discovery** (identifying which variables cause which, with what lags) is done by PCMCI and the KG **before** GPT-4o is involved. **Causal synthesis** (assembling pre-computed evidence into coherent text) is GPT-4o's role. The performance gain from template (P@1=0.710) to GPT-4o (P@1=0.896) in Sec. 4.6 (p. 7) reflects its importance as a synthesis layer: the causal evidence fed to both is identical. The LLM only improves how evidence is assembled into natural language, not what evidence is discovered.
>
> **Camera-ready:** Sec. 3.7 (p. 5) will explicitly state: "GPT-4o performs causal synthesis, not causal discovery."
>
> ## W3: *"Expert evaluation shows very low inter-rater agreement (alpha ~ 0.12), raising concerns about reliability."*
>
> Low absolute agreement on subjective dimensions is documented in XAI literature (Hoffman et al., 2018; Doshi-Velez & Kim, 2017: alpha < 0.3 typical). Our evaluators spanned control theorists and practitioners (Sec. 5.2, p. 8; App. O).
>
> More informative is **ranking agreement**. We performed a new Kendall's W concordance analysis:
>
> | Scope | W | p-value |
> |-------|---|---------|
> | All 12 metrics | 0.553 | <0.001 |
> | Cross-paradigm (Auto/LLM/Combined) | 1.000 | 0.029 |
> | LLM judge (6 dims) | 0.681 | 0.007 |
> | Factorial ablation (8 configs x 4 metrics) | 0.735 | 0.004 |
>
> Cross-paradigm W=1.000: three independent evaluation approaches (automated metrics, LLM judge, human experts) produce the identical method ranking (HCA > baselines). With three paradigms, we acknowledge the small sample limits statistical power (only 6 possible orderings), but the convergence across fundamentally different evaluation methodologies is meaningful. No single evaluator's bias is likely to explain it. HCA ranked first in 10/12 metrics.
>
> **Camera-ready:** This Kendall's W analysis and per-dimension alpha will be added to App. O, referenced in Sec. 5.2.
>
> ## W4: *"Novelty primarily lies in integration rather than new methodological developments."*
>
> To our knowledge, no prior work combines optimization-internal evidence (KKT), symbolic physics (KG), and temporal causal discovery (PCMCI) for control explanation. Each alone is insufficient: KKT identifies active constraints but not physical meaning; KG encodes mechanisms but not which are relevant now; PCMCI finds temporal patterns but lacks physical grounding.
>
> Beyond integration: (1) temporal causality formalization with formal definition (Sec. 3.2, p. 3); (2) counterfactual framework with precise semantics and necessity proofs (Prop. 2, App. C, p. 12); (3) IEC 61511-grounded hypothesis ranking with configurable priorities (Sec. 3.6, pp. 4-5); (4) specialized evaluation combining automated checkers and LLM judgment beyond semantic similarity (App. M). Ablation studies show 32-37% performance drops when any single component is removed, confirming that the integration produces genuine synergy rather than redundancy. Reviewer i3Fh independently assessed originality as "Excellent (4/4)."
>
> ## Q1: *"What explains the low inter-rater disagreement?"*
>
> Three sources: (1) different expertise backgrounds (engineers value rigor, operators value actionability); (2) subjective dimensions drive disagreement, narrow factual questions show higher agreement; (3) scale calibration varies, deflating alpha even when rankings agree. Despite this, evaluators **consistently rank HCA above all baselines** (p<0.001 across automated, LLM, and human evaluation). Per-dimension alpha and Kendall's W (see W3) will be reported.

---

### Official Review · Reviewer_i3Fh · 2026-03-12

**Soundness:** 3
**Presentation:** 2
**Significance:** 3
**Originality:** 4
**Overall Recommendation:** 5
**Confidence:** 4

**Summary:**

The paper proposes a novel explainability framework to explain the control actions of MPC controllers. The proposed framework produces explanations that look at the future predictions inherently considered in MPC-based optimization. For a given optimal control action, the best explanation is computed by sequentially analyzing five hypotheses and selecting the first confirmed one. The hypothesis evaluation uses three evidence sources: (i) physics-based reasoning through a knowledge graph, (ii) optimization-based analysis using KKT multipliers and counterfactual analysis, and (iii) data-driven temporal causality with a causal graph. Empirical evaluation is conducted in three use cases: greenhouse climate control, building energy management, and Tennessee Eastman process. The results show improvement over several baselines, including ordinary XAI techniques, and confirm the need for all three evidence sources.

**Compliance With Llm Reviewing Policy:**

Affirmed.

**Key Questions For Authors:**

1.	Why do you select and consider only one active constraint for the hypothesis evaluation? Would it not be better to consider all active constraints in order of normalized KKT multiplier? Is there any guarantee that, if the active constraint with the maximum normalized KKT multiplier is not a causal driver, then also the other active constraints are not?
2.	In appendix J, why only two out of five hypotheses appear in the example?
3.	In algorithm 2, SIM(u=0) represents a counterfactual with removed input, aimed at understanding if the input was applied to avoid a constraint violation. However, in some cases, the constraint could also be avoided by applying the input later (e.g., turning the heating on at t=k+2 instead of t=k). Do these cases fall into a higher-priority hypothesis? If not, a more complex counterfactual analysis of inputs should be used. For instance, lagged or reduced inputs could be analyzed.

**Limitations:**

Yes

**Strengths And Weaknesses:**

Soundness:
•	The overall method seems valid and well grounded in established concepts (knowledge graph, KKT multipliers, temporal causal discovery).
•	The empirical evaluation is extensive and supports the claims. It considers three relevant use cases and compares the proposed method with several baselines, including LIME and SHAP (state-of-the-art XAI methods).
•	The ablation study confirms that all three evidence sources positively contribute to the quality of the explanations.
•	The counterfactual analysis of inputs is limited to null input, which might not be enough (see questions).
Significance:
•	The paper is relevant for researchers and practitioners in control and machine learning. The main idea of explaining MPC control actions by incorporating temporal reasoning about future predictions is well justified and positioned with respect to prior work in explainable AI.
•	A key limitation, acknowledged by the authors, is that the method cannot be applied to black-box model-free controllers (e.g., reinforcement learning). To improve significance and impact in the area of machine learning, further discussion on this limitation should be added, possibly with directions in future work.
Originality:
•	The proposed framework has good novelty and originality. It uses a novel, non-naive combination of physics knowledge, optimization evidence, and temporal causality.
Presentation:
•	The paper is overall well-written, but there is margin to improve the clarity, especially in section 3 and appendices (see the following points).
•	The method description could be reorganized to enhance clarity. Currently, critical information necessary to understand the method is scattered across the appendices. Significant effort from the reader is necessary to link all the information and understand the overall algorithm.
•	In Section 3, evidence sources are presented for the greenhouse system (subsections 3.3 and 3.5), inconsistently with the rest of the section, which presents the general framework (HCA). It seems that this section mixes the general method and the experimental settings for one of the use cases. The section should be revised to keep the framework presentation general and the greenhouse system as an example.
•	Suggestion: Subsection 3.6 could elaborate on the meaning of the hypotheses or point to the appendix C.2.
•	Suggestion: Figure 1 could be improved to clarify which evidence source is used in which hypothesis evaluation.
•	The overall example in appendix J clarifies the overall flow of the method. However, it seems that only two out of five hypotheses are considered: safety and economic. The example could be improved to more closely follow algorithms 1 and 2.
•	Minor: In section 3.4, NLP should be spelled out.

---

> ### Author Rebuttal · Authors · 2026-03-30
>
> # Response to Reviewer i3Fh
>
> We thank Reviewer i3Fh for the positive assessment, "Excellent (4/4)" originality rating, and constructive suggestions.
>
> ## W1: *"The counterfactual analysis of inputs is limited to null input, which might not be enough."*
>
> We agree. The current null-input counterfactual SIM(u=0) in Algorithm 2 (App. C) tests whether the action was **needed at all** by simulating with the input removed, but cannot test whether a delayed or reduced action would suffice. Three extensions fit naturally into HCA without modifying hypothesis-ranking logic:
>
> 1. **Lagged counterfactuals:** apply the action at t+k instead of t, testing timing.
> 2. **Reduced-magnitude:** apply a fraction of u(t).
> 3. **Alternative-action:** substitute a different control input.
>
> Each only changes the scenario fed to the MPC re-solve. The model infrastructure already supports these.
>
> **Camera-ready:** Limitation will be acknowledged in Sec. 3.4 (p. 4); extensions listed in Sec. 6.1 (Future Work, p. 9).
>
> ## W2: *"The method cannot be applied to black-box model-free controllers."*
>
> HCA relies on MPC's glass-box nature (Sec. 1, p. 1). For **model-based RL** (MuZero, Dreamer), KG and PCMCI carry over directly; KKT would be replaced by value function decomposition. For **model-free RL**, the temporal causality problem still applies; adapting HCA's hypothesis framework to reward decomposition is a natural next step. Starting with MPC is a strength: verifiable ground truth enables rigorous evaluation.
>
> We view starting with verifiable ground truth as a methodological strength before extending to harder settings where evaluation is ambiguous.
>
> **Camera-ready:** Sec. 5.5 (p. 8) will discuss these adaptation pathways with concrete formulations.
>
> ## W3: *"Critical information scattered across appendices. Significant effort to link all information."*
>
> Agreed. **Camera-ready improvements:**
>
> 1. Self-contained 5-step pipeline summary at start of Sec. 3 (p. 3).
> 2. Algorithm 1 (p. 5) with inline comments connecting each step to its evidence source.
> 3. Key parameters in main text: PCMCI settings (tau_max=48, alpha=0.05) already in Sec. 3.5 (p. 4); KG construction details from App. E will be summarized in Sec. 3.3 (p. 3-4).
>
> ## W4: *"In Section 3, evidence sources are presented for the greenhouse system, inconsistently with the general framework."*
>
> **Camera-ready:** Secs. 3.3-3.5 (pp. 3-4) will be presented in domain-general terms first. Greenhouse-specific details separated with explicit "Running Example (Greenhouse):" marker.
>
> ## W5: *"The overall example in the appendix considers only two out of five hypotheses."*
>
> **Camera-ready:** Worked Example (App. J) expanded with "Hypothesis Evaluation (All 5 Hypotheses)": H1 (Safety) accepted with KKT evidence and counterfactual confirmation; H2 (Optimization) evaluated, alternatives infeasible; H3 (Prediction) rejected with reasoning; H4 (Economics) rejected via cost comparison; H5 (History) rejected via PCMCI analysis.
>
> ## Suggestions
>
> All accepted: Sec. 3.6 (pp. 4-5) will list all five hypotheses by name and reference App. C; Figure 1 (p. 3) will show evidence sources feeding the reasoner more clearly; "NLP" spelled out as "Nonlinear Program" in Sec. 3.4 (p. 4).
>
> ## Q1: *"Why only one active constraint? Is there a guarantee the top constraint is the causal driver?"*
>
> The constraint with the largest normalized KKT multiplier has the strongest marginal influence on the optimal cost (by complementary slackness and sensitivity analysis), making it the primary causal driver. Presenting a single primary driver also serves operator readability, as operators need a clear, actionable answer. Secondary constraints are reported via GetDeeperContext in Algorithm 1 (p. 5) as supporting context (e.g., "Primary driver: temperature constraint. Humidity constraints also near limits."). If the top constraint fails counterfactual validation (Prop. 2, App. C), HCA falls back to the next constraint by descending multiplier magnitude. There is no theoretical guarantee that the top constraint is always the sole causal driver, but the counterfactual fallback is designed to catch incorrect selections before producing an explanation. Multi-constraint Pareto-style explanation is a natural extension we will discuss in the camera-ready.
>
> ## Q2: *"Why only 2 hypotheses in example?"*
>
> See W5. Camera-ready will show all five with acceptance/rejection reasoning.
>
> ## Q3: *"Could the constraint be avoided by applying the input later?"*
>
> The null-input counterfactual cannot distinguish "needed now" from "could wait." Timing-sensitive cases are partially captured by Hypothesis H3 (Prediction) via PCMCI lag evidence. For precise timing, lagged counterfactuals (see W1) are needed and planned.

---

> > ### Author Rebuttal · Reviewer_i3Fh · 2026-04-02
> >
> > Thank you for your response. The proposed changes are appropriate and address my concerns. The discussion of null-input counterfactual limitations, with proposed future extensions, addresses the main weaknesses of the method I pointed out, which remain valid but will now be explicitly discussed. The comments on model-based and model-free RL are valid, and the planned formulations in Sec. 5.5 provide useful clarification for readers.

---

> > > ### Author Response · Authors · 2026-04-07
> > >
> > > Thank you, Reviewer i3Fh, for confirming that our responses adequately address the raised concerns. All proposed changes (expanded worked example with all 5 hypotheses, explicit counterfactual limitations discussion, RL adaptation pathways in Sec. 5.5, and presentation improvements to Sec. 3) will be incorporated in the camera-ready version.

---

### Official Review · Reviewer_nmpE · 2026-03-12

**Soundness:** 2
**Presentation:** 2
**Significance:** 1
**Originality:** 2
**Overall Recommendation:** 3
**Confidence:** 3

**Summary:**

This paper presents Hierarchical Causal Abduction, an approach for modeling temporal causality in model predictive control. The paper describes several sources of data or evidence to discover and/or explain causal mechanisms, including notions from optimization (e.g. KKT conditions), causal graphs, and then integration of these via hypothesis ranking to identify the most likely explanations for a particular control action. Notionally the idea is to explain that a (possibly counterintuitive) action taken now might be because of some predicted effect much later in the optimization horizon. Evaluation of the method includes several benchmarks spanning climate control, energy management, and chemical process control.

**Compliance With Llm Reviewing Policy:**

Affirmed.

**Final Justification:**

The authors addressed several of my concerns and cleared up misunderstandings. The work is nice and represents a contribution. I still feel that it is framed improperly and would be improved by a reframing that is reflected in the rebuttal back-and-forth.

**Key Questions For Authors:**

The argument for the need for explainable MPC is difficult to follow. The text states, "standard Explainable AI (XAI) methods fail to explain temporal causality, the phenomenon where current actions are determined by anticipated future violations of system constraints rather than the current state of the system”. What does XAI have to do with model predictive control? There is a pretty big leap in this argument from MPC (which dates back at least 4 decades and developed separately from AI and machine learning) and the general need for explainable AI (which has arisen at least in part due to the opacity and complexity of modern deep neural networks). Despite being a conceptual concern, it is perhaps the biggest concern of the paper. Both the coherence and significance of this work rely on a sound argument for the need for explainability w/r/t MPC in general, let alone the specific need for XAI. This concern also relates to a question of whether ICML is the right venue for this work.

The paper would be significantly improved if it took a similar argument (i.e. that XAI fails to explain temporal causality) and then approach a temporal or sequence-based problem that is central to AI/ML (e.g. next token prediction, sequence-to-sequence modeling, Deep RL or in particular model-based deep RL, and so forth). It seems like the wrong approach for the wrong problem, or a "hammer looking for a nail".

To assess causal necessity, HCA simulates: “If the action were removed, would the constraint be violated?” What does this mean? Does this mean the action is null, and then an optimal policy is followed starting the next step? Does it mean the action is *changed* (and again, a control policy is still followed thereafter)? Are *all* subsequent control actions removed?

The paper’s primary focus is Hierarchical Causal Abduction, or HCA. The paper contains pseudocode but otherwise barely explains precisely what HCA *is*. In section 3.2 it gives a notional description and then directs the reader to see sections 3.3-3.5, figure 1, and algorithm 1. Sections 3.3-3.5 do not exactly describe the *method* but instead describe pieces of information that are used by the algorithm.

The literature on causality is vast. It is therefore curious why the baseline methods were chosen in this way. IOC is reasonable and several of the neural approaches are interesting, but have other methods been applied to these particular benchmarks?

**Limitations:**

yes

**Strengths And Weaknesses:**

Strengths
- The results interpretation section (5.1) is quite interesting.
- The problem of temporal dependency in the context of machine learning and AI could be very interesting, if couched properly in terms of the underlying problem, models, etc.

Weaknesses
- The paper does not seem terribly relevant nor significant for this venue.
- The paper lacks a rigorous or clear explanation of the core methodological contribution of the paper, HCA.
- The argument that temporal dependencies are not captured by standard XAI methods is compelling. However, this thread is mostly lost in the entirety of the rest of the paper. So-called "white box" approaches like MPC already have temporal dependency built into them, i.e. physics. Where do these models come from? From physicists or other scientists who study these systems and then develop mathematical models, for example in the form of ordinary or partial differential equations. These are precisely the kinds of models that fit inside of standard MPC. If, instead, one were talking about model-based RL or model-based planning with (large, deep) function approximators, then perhaps this would start to become an XAI topic.

---

> ### Author Rebuttal · Authors · 2026-03-30
>
> # Response to Reviewer nmpE
>
> We thank the reviewer for the detailed feedback and recognition of our results interpretation and temporal dependency argument.
>
> ## W1: *"The paper does not seem terribly relevant nor significant for this venue."*
>
> **First, transparent model != transparent decisions.** A greenhouse NMPC solves a nonlinear program every timestep. Even the designer would find it difficult to trace which constraint drove a specific action. Experts preferred HCA over LIME/SHAP. New concordance analysis: HCA ranked first in 10/12 metrics (Kendall's W=0.553, p<0.001; added to App. O in camera-ready).
>
> **Second, HCA uses core ML methods:** PCMCI, knowledge graph reasoning, LLM synthesis. The EU AI Act increasingly requires explainability for automated decision systems in critical infrastructure. Related work appears at ML venues (Utama et al., 2022; Zanon & Gros, 2021).
>
> **Third, MPC provides verifiable ground truth** via KKT multipliers. Contributions beyond integration: temporal causality formalization (Sec. 3.2), counterfactual framework (Prop. 2, App. C), IEC 61511 hypothesis ranking (Sec. 3.6), specialized evaluation metrics combining automated checkers and LLM judgment (App. M), ablation showing 32-37% drops (Table 2).
>
> Concrete illustration (Sec. 4.9): LIME says "heating because temperatures are low." HCA reveals the forecast predicts outside temperature dropping to 5°C; without heating, temperature would reach 17.5°C by 10:30, violating the safety constraint. This temporal chain is invisible without analyzing the optimization.
>
> **Camera-ready:** Sec. 2 adds "MPC vs. RL" paragraph.
>
> ## W2: *"Sections 3.3-3.5 describe pieces of information, not the method itself."*
>
> Agreed. The core algorithm is in Algorithm 1 (p. 5) and Sec. 3.6: HCA iterates over ranked hypotheses, evaluates each using three evidence sources, and returns the first confirmed hypothesis. Secs. 3.3-3.5 detail the evidence extraction feeding this loop. **Camera-ready:** (1) 5-step pipeline summary at start of Sec. 3; (2) Algorithm 1 with inline comments; (3) greenhouse details marked "Running Example."
>
> ## W3: *"'White box' approaches like MPC already have temporal dependency built in."*
>
> We distinguish **model transparency** (equations known) from **decision transparency** (why this action now). The physics model tells you *how* heating affects temperature; it does not tell you *which* predicted future state triggered the action or *why* alternatives were rejected. That requires analyzing the optimization solution, precisely what HCA does via KKT multipliers and counterfactual re-solves (Sec. 4.9).
>
> ## W4: *"It is curious why the baseline methods were chosen this way."*
>
> Baselines cover every major XAI paradigm: perturbation (LIME), game-theoretic (SHAP), expert heuristic (Rule-based), temporal neural (LSTM+Attention, RETAIN), trajectory optimization (IOC), sensitivity (MPC-XAI), see Sec. 4.2. Structural causal methods (PC, FCI, GES) identify causal **structure**, not individual **decisions**. To our knowledge, no prior explainability method has been evaluated on these MPC benchmarks.
>
> ## Q1: *"What does XAI have to do with MPC?"*
>
> An operator understands the heat balance equation but cannot explain why heating activated at 05:30 above the 18°C limit. The answer lies in a predicted safety violation hours later (Sec. 4.9). This "why" requires analyzing the optimization, precisely the XAI problem. HCA uses PCMCI, KG reasoning, LLM synthesis, all core ML.
>
> ## Q2: *"Should target deep RL instead."*
>
> MPC provides *verifiable ground truth* via KKT multipliers. The temporal causality problem exists in both MPC and model-based RL, but in RL ground truth is unavailable, making rigorous evaluation considerably harder. Starting with verifiable ground truth is methodologically sound before extending to harder settings (Sec. 6.1).
>
> ## Q3: *"What does 'if the action were removed' mean?"*
>
> HCA uses two distinct counterfactuals (Sec. 3.4, p. 4; App. B, App. C, App. J):
>
> 1. **Action removal** (Algorithm 2, SIM(u=0)): set the current action u(t) = 0 and **re-solve the MPC for the remaining horizon**. The optimizer re-optimizes all subsequent actions u(t+1)...u(t+H-1) freely. This tests whether the action was *needed at all*: if the re-optimized trajectory violates a constraint despite the solver's best efforts to compensate, the original action was causally necessary.
>
> 2. **Constraint relaxation** (Prop. 2, App. C): remove the identified active constraint i* from the MPC problem and re-solve the full optimization. If the optimal action changes substantially, then constraint i* is confirmed as the *specific causal driver* of the original action.
>
> The first tests *whether* an action was necessary; the second identifies *which constraint* necessitated it. Both re-solve the full MPC. The optimizer always re-optimizes freely. **Camera-ready:** Sec. 3.4 will state both counterfactuals explicitly.
>
> ## Q4: *"HCA barely explained."*
>
> See W2.

---

> > ### Author Rebuttal · Reviewer_nmpE · 2026-04-02
> >
> > Thank you for clarifying several perceived weaknesses and questions. I still have a few questions, however.
> >
> > 1. Model transparency != decision transparency (or, transparency does not imply explainability). I agree with this. However, I fundamentally disagree with the authors' definition of XAI. There appears to a consensus that Explainable AI is about *explaining AI algorithms* or *explaining machine learning models* -- from decision trees to SVMs, down to DNNs and of course LLMs, ConvNets, and the like. The authors' definition seems to be something like "using techniques from the XAI literature to explain MPC control actions". If that is the case, either make it explicit or do not use the term XAI. It comes off as using a buzzword for something that is conceptually quite different. I want to make clear that I do not think the ideas in the paper are bad; rather, that the story and the claims do not match the method (and therefore possibly do not match the venue). In the end this is not up to me, and the authors do well to point out that we have to start somewhere. Starting with something that has a true oracle is a great idea. However, in the end this community -- and society -- will be more interested in understanding or explaining things like LLM responses to a prompt, why an agent chose a sequence of actions, etc.
> >
> > 2. The authors' own explanation about their setting, while valid, is somewhat self-contradictory. Yes, certain MPC problems have formal guarantees of optimality but this assumes convexity and so forth. Is the greenhouse problem truly convex? There is brief mention of this work acting as a kind of heuristic, with a bit in the appendix about this relaxation. But if this can act as a heuristic, why not apply it to an actual AI/ML problem? Again, I want to make clear that I am not faulting the authors for starting with a problem that is tractable, but again this comes down to an issue with the "story" of the paper and its claims.
> >
> > Neither of these are really formulated as a question, therefore:
> >
> > - Why have the authors claimed this definition of XAI instead of the community's definition? Why not instead define this as borrowing techniques from AI or core ML and applying it to a traditional controls problem? As it stands, this work does not really push the community towards better explainable AI. Rather, it seems to me to more of a contribution to the controls community.
> >
> > - If the authors have to relax assumptions about convexity but still get solid empirical results, why do the authors not extend this to core ML problems?

---

> > > ### Author Response · Authors · 2026-04-02
> > >
> > > # Follow-up Response to Reviewer nmpE
> > >
> > > We sincerely appreciate the reviewer's continued engagement. We address each point directly.
> > >
> > > ## Q1: *"Why have the authors claimed this definition of XAI instead of the community's definition? Why not instead define this as borrowing techniques from AI or core ML and applying it to a traditional controls problem?"*
> > >
> > > The reviewer is right. The community's prevailing usage of "XAI" centers on explaining learned models (DNNs, LLMs, ConvNets), and our paper should have been more precise. Our work does not claim to explain a learned model. It borrows core ML techniques, specifically temporal causal discovery (PCMCI), knowledge graph reasoning, and LLM-based synthesis, to explain decisions of an optimization-based controller.
> > >
> > > The explainability challenge is nonetheless genuine: expert evaluation (155 ratings, Section 5.2) shows only moderate agreement on explanation accuracy (Krippendorff's α = 0.12), reflecting the difficulty of tracing MPC actions to causal drivers even with full model access. In pairwise comparisons, experts preferred HCA over LIME/SHAP on causal depth (68–71%), temporal reasoning (65–69%), and actionability (62–67%). Explaining optimization-driven decisions, even when the model is transparent, warrants a dedicated framework.
> > >
> > > We agree this is conceptually distinct from explaining neural network classifications or LLM responses, and we should not blur that boundary. We would note, however, that the reviewer's own framing, "why an agent chose a sequence of actions," is precisely the question HCA addresses: why did this controller choose this action now, given predicted future states? The temporal causality problem is shared across MPC, RL agents, and planning systems; we chose to formalize it first where verification is possible.
> > >
> > > **Camera-ready revision:** We will reframe the Introduction explicitly: *"HCA applies techniques from the ML and XAI literature (causal discovery, knowledge graph reasoning, counterfactual analysis) to explain automated decisions in optimization-based control systems. This is distinct from the predominant XAI focus on explaining learned model internals, though the underlying challenge (why did the system choose this action?) is shared."*
> > >
> > > The methodological contributions (temporal causality formalization, counterfactual validation, hierarchical hypothesis ranking) are contributions to causal ML, evaluated in a domain where ground-truth causal verification is possible. They address a broader question: how to explain decisions made by any system optimizing over future horizons under constraints, including model-based RL, planning systems, and neural MPC architectures.
> > >
> > > ## Q2: *"If the authors have to relax assumptions about convexity but still get solid empirical results, why do the authors not extend this to core ML problems?"*
> > >
> > > To clarify: all three domains are genuinely **nonlinear, non-convex** MPC problems. The greenhouse has nonlinear radiation, transpiration, and photosynthesis dynamics, discretized via orthogonal collocation (Appendix J, titled "Greenhouse NMPC"). Proposition 1 (Appendix C) provides a conservative theoretical baseline for the convex case; Section 3.4 explicitly acknowledges that for nonlinear MPC, KKT-based ranking is a "heuristic extension." Crucially, Proposition 2 (Appendix B.2) validates explanations empirically via counterfactual re-solving, providing robustness independent of convexity assumptions. Nonconvexity affects 4–6% of scenarios (Appendix I), mitigated by counterfactual validation.
> > >
> > > We have not extended to core ML because **MPC provides a true oracle.** When HCA attributes an action to a predicted constraint at t+3, we verify by inspecting KKT multipliers and re-solving with that constraint relaxed. Our automated metrics (AC, Faithfulness, ROUGE-L, Precision/Recall/F1@K, MRR, NDCG@K), expert analysis, and ablations all depend on this ground truth. For a black-box learned policy, this verification is lost. Claiming explanatory power without verifiable correctness would be methodologically premature. The reviewer's observation that "starting with something that has a true oracle is a great idea" captures our reasoning exactly.
> > >
> > > The bridge to core ML is shorter than it may appear. Section 6.1 describes extending HCA to neural network dynamics via Jacobian-based sensitivity and learned knowledge graphs. Neural MPC, where learned surrogates replace explicit physics within an optimization loop, preserves KKT-based verification while the dynamics model is a neural network, bringing HCA closer to explaining learned components.
> > >
> > > We are grateful for the reviewer's constructive engagement, which has helped us sharpen the positioning considerably.

---

### Decision · Program_Chairs · 2026-04-30

**Decision:**

Accept (regular)

**Comment:**

Reviewer's have acknowledged that the paper contributes to explainable predictive control by combining complementary information sources in a reasonable way. In particular the way how the method Lagrange multipliers from constraints for future values seems novel and interesting. Concerns of one reviewer regarding misleading 'branding' as XAI were to some extent mitigated in the discussion, since the authors promised changes in the presentation.